biomaterials/bioengineering/biophysics

bone, stress relaxation, relaxation modulus, Kohlrausch–Williams–Watts function, viscoelasticity

**Author for correspondence:**
Saulo Martelli
e-mail: saulo.martelli@qut.edu.au

# The effect of age and initial compression on the force relaxation response of the femur in elderly women

## Saulo Martelli[1,2]

[1]Centre for Biomedical Technologies, School of Mechanical, Medical and Process Engineering, Queensland University of Technology (QUT), Brisbane, Australia
[2]Medical Device Research Institute, College of Science and Engineering, Flinders University, Tonsley SA, Australia

SM, 0000-0002-0012-8122

The effect of force amount, age, body weight and bone mineral density (BMD) on the femur's force relaxation response was analysed for 12 donors (age: 56–91 years). BMD and fracture load, $F_L$, were estimated from clinical CT images. The 30 min force relaxation was obtained using a constant compression generating an initial force $F_0$ between 7% and 78% of $F_L$. The stretched decay function $(F(t) = A \times e^{(-t/\tau)\beta})$ proposed earlier for bone tissue was fitted to the data and analysed using robust linear regression. The relaxation function fitted well to all the recordings $(R^2 = 0.99)$. The relative initial force was bilinearly associated $(R^2 = 0.83)$ to the shape factor, $\beta$, and the characteristic time, $\tau$, when $F_0/F_L$ was less than 0.4, although $\beta$ was no longer associated with $F_0/F_L$ by pooling all the data. The characteristic time $\tau$ increased with age $(R^2 = 0.37, p = 0.03)$ explaining 35% of the variation of $\tau$ in the entire dataset. In conclusion, the relative initial force mostly determines the femur's force relaxation response, although the early relaxation response under subcritical loading is variable, possibly due to damage occurring at subcritical loading levels.

## 1. Introduction

Bone exhibits a dynamic mechanical and biological response to loading [1–6]. Bone stiffness, ultimate strength, energy to failure, toughness, fatigue life and crack growth are all loading rate-dependent [1,2,6] and so is bone metabolism, which adapts bone architecture to changes in time of loading amount and frequency [3–5]. For example, a 10-fold strain rate increase from $0.001 \text{ s}^{-1}$ (physiological) to $0.01 \text{ s}^{-1}$ causes an 18–19% increase of the ultimate compressive strength and energy absorption

capacity as the elastic modulus increases by 7% [1,6]. Similarly, bone metabolism adapts bone architecture to changes of strain and strain rate by balancing osteoclastic resorption and osteoblastic bone deposition processes [5]. Yet, the dynamic properties of entire bones are not fully elucidated.

Stress relaxation is a dynamic property concerning the stress decline over time in material under constant deformation. Two concomitant relaxation processes were found in isolated bone cores [7–9]. The fast process, thought to be governed by the relative motion of collagen molecules [7,8], explains the large majority of the stress decline over time and can be described by a stretched exponential decay function with a characteristic time equal to 25–45 min [9]. The slow process, thought to be governed by a viscous-like motion between cement lines [7,8], can be described by an exponential decay function with a characteristic time equal to 70–300 h [9]. Age and gender have been shown to modulate bone viscoelasticity by affecting cross-link concentration [10,11] and the interface between mineral and collagen phases [12,13]. However, relating the stress relaxation observed in isolated bone cores and the force relaxation response of whole bone organs remains difficult.

In entire bone organs, several complementary mechanisms can modulate the two relaxation mechanisms observed in isolated bone cores. For example, the superposition of relaxing stresses of different amplitudes in the spatially heterogeneous structure of entire bones, the marrow flow through the bone pores (poroelasticity), and damage are three different mechanisms potentially providing bone organs with a specific relaxation response otherwise not observable in isolated bone cores [13–15]. In the human femur, creep was found to cause cortical strain changes smaller than 3% after 30 s from load application [16] and a 10-fold strain rate increase causes cortical strain and bone stiffness changes smaller than 8% [17]. No significant changes were observed for the tibia and the fibula [17]. To the best of the author's knowledge, no study has yet investigated the force relaxation response in the proximal human femur.

The aim of the present study was (i) to test the suitability of earlier bone stress relaxation theories for predicting the force relaxation response of human femora and (ii) to falsify the hypothesis that ageing modulates the femur's relaxation response. The femur's force relaxation response was obtained from 12 elderly female donors between 56 and 91 years of age. Thirty force relaxation profiles spanning an initial compression from moderate to subcritical were obtained by pooling published data and novel measurements obtained with two established testing protocols. The stretched exponential relaxation function by Sasaki and co-workers [9] was fitted to the data. The relationship between force relaxation, age, body weight, bone mineral content and density was studied using robust linear regression analysis.

# 2. Methods

Ethics clearance (Project no. 6380) was obtained from the institutional Social and Behavioural Research Ethics Committee (SBREC). The force relaxation profiles were obtained using two different custom-made compression devices during two different experimental campaigns with four specimens [18,19] and here extended to 12 specimens under a range of initial compression levels from moderate to subcritical. One device applied and kept constant over time a controlled compression causing a reaction force along a prescribed direction (8° abduction) within a compressive chamber suitable for concomitant microstructural imaging at the Australian Synchrotron (Clayton VIC, Australia) [18]. The second device was a custom-made six-degree-of-freedom robot used to apply to the specimen assembly, and keep constant over time, a displacement generating a reaction force along the prescribed direction. The first device was used to extend the force relaxation profiles reported earlier [18,20] for four specimens to 12 specimens in total using an initial compression equal or above one-fourth of the estimated fracture load. The second device was used to provide additional force relaxation profiles using an initial compression below one-fourth of the estimated fracture load. The effect of the different testing protocols and of the non-bone components of the specimen assembly was assessed by monitoring the deformation of the bone and comparing the force profiles.

## 2.1. Specimens

Twelve femora were obtained from elderly female donors through a dedicated body donation programme (Science Care Inc., Phoenix, USA) (table 1) who collected the written informed consent from the legally authorized representative or next of kin. Exclusion criteria included possible bone metastases and any other disease known to impede normal ambulation up to 1 year before death. The primary cause of death included septic shock, dysphagia, cerebral vascular failure, renal failure,

**Table 1.** Body anthropometry and bone quality descriptive statistics.

| spec. ID | height (cm) | weight (kg) | age (years) | body mass index (BMI) | bone mass (g) | area (cm$^2$) | BMD (g cm$^{-2}$) | osteoporosis level (T-score) | fracture load (N) |
|---|---|---|---|---|---|---|---|---|---|
| 1 | 170 | 58 | 79 | 20 | 1.81 | 3.92 | 0.46 | −3.0 | 1998 |
| 2 | 145 | 104 | 80 | 50 | 0.97 | 2.33 | 0.42 | −3.4 | 3788 |
| 3 | 142 | 57 | 91 | 28 | 1.59 | 3.52 | 0.45 | −3.1 | 4075 |
| 4 | 168 | 143 | 66 | 51 | 1.69 | 3.05 | 0.56 | −2.1 | 7044 |
| 5 | 155 | 59 | 76 | 25 | 1.72 | 3.11 | 0.55 | −2.1 | 5246 |
| 6 | 155 | 136 | 70 | 57 | 2.11 | 3.49 | 0.60 | −1.6 | 5956 |
| 7 | 163 | 116 | 56 | 44 | 2.53 | 2.98 | 0.85 | 0.8 | 8636 |
| 8 | 152 | 88 | 78 | 38 | 1.16 | 2.90 | 0.40 | −3.6 | 4742 |
| 9 | 150 | 32 | 77 | 14 | 1.24 | 2.86 | 0.43 | −3.3 | 2641 |
| 10 | 178 | 91 | 75 | 29 | 0.84 | 2.98 | 0.28 | −4.7 | 3306 |
| 11 | 163 | 118 | 68 | 45 | 1.78 | 3.65 | 0.49 | −2.7 | 4518 |
| 12 | 168 | 78 | 81 | 28 | 1.68 | 2.53 | 0.66 | −1.0 | 6128 |
| min | 142 | 32 | 56 | 14 | 0.84 | 2.33 | 0.28 | −4.7 | 1998 |
| max | 178 | 143 | 91 | 57 | 2.53 | 3.92 | 0.85 | 0.8 | 8636 |

ventricular fibrillation, hypertension, arteriosclerosis, chronic obstructive pulmonary disease and Alzheimer. Samples were stored at −20°C at the Biomechanics and Implants Laboratory (BIL) of Flinders University. Before testing, specimens were thawed at room temperature for approximately 10 h while wrapped in fabric tissue soaked with phosphate-buffered saline solution to maintain bone moisture.

## 2.2. Bone quality

Specimens were scanned using a clinical CT scanner (Optima CT660, General Electric Medical Systems Co., Wisconsin, USA). The helical scanning protocol (tube current: 300 mA; voltage: 140 kVp) provided a 0.7 mm slice thickness and 0.63 mm in-plane pixel size. A phantom (Mindways Software, Inc., Austin, USA) with five samples of known dipotassium hydrogen phosphate density ($K_2HPO_4$ equivalent density range: 51.8–375.8 mg cm$^{-3}$) was scanned with the samples. The fracture load was calculated using a previously validated ($R^2 = 0.89$) finite-element procedure [18,21]. In summary, the CT images were calibrated to equivalent bone mineral density levels in the phantom. The femur geometry was extracted from the CT images using a semi-automatic segmentation procedure (Simpleware, Exeter, UK). The voxels in the volume of images were converted into a mesh of quadratic hexahedral elements. Locally isotropic material properties were defined using the density-to-elastic modulus relationship by Schileo *et al*. [21]. The model was subjected to a nominal force of 1000 N mimicking the single-leg stance load configuration used for testing. The nodal strain in the model was averaged over a 3 mm diameter spherical volume. The fracture load was estimated by scaling the nominal load to match the yield strain, either in tension (0.73%) or in compression (1.04%) [22]. The total hip area, bone mineral content (BMC), bone mineral density (BMD) and the corresponding osteoporosis level (T-score) were estimated from the CT mages by following the guidelines by Khoo *et al*. [23].

## 2.3. Specimen preparation

The femoral diaphysis was cut 180 mm distally from the proximal femoral head and potted 55 mm deep in aluminium cups using dental cement (Soesterberg, The Netherlands), which met the ISO 5833 requirements. The femoral head centre was aligned with the vertical axis of the cup. The longitudinal axis of the femoral diaphysis was oriented at 8° in the frontal plane so that the vertical axis of the cup mimicked the force orientation during a static single-leg stance task [24] (figure 1; electronic supplementary material, figure S1).

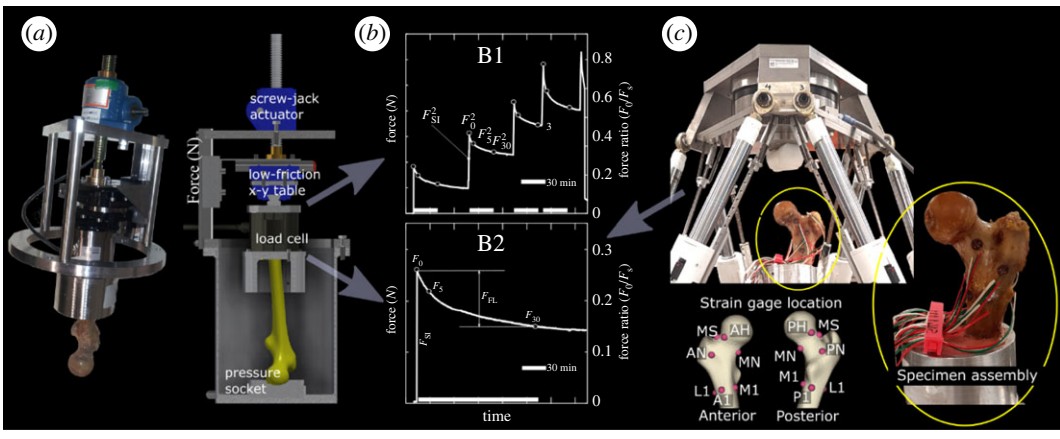

**Figure 1.** On the left-hand side (*a*), a photo of the upper part of the compressive stage for the microstructural imaging experiment. A photo of the upper part of the device and a section of the three-dimensional model of the whole compressive stage are displayed. In the (*b*), a representative force profile to fracture (B1), and a single loading cycle (B2). The compressive force at $t_0$ ($F_0$), at 5 and 30 min relaxation ($F_5$ and $F_{30}$), and the force step increment ($F_{SI}$) are also indicated. On the right-hand side (*c*), the robot testing device. The four strain gauges in the anterior (A1), posterior (P1), medial (M1) and lateral (L1) aspect, the three strain gauges in the anterior (AN), posterior (PN) and medial (MN) aspects of the femoral neck, those in the anterior (AH) and posterior (PH) aspect of the sub-capital head and the strain gauge in the mid-superior (MS) femoral neck are also schematically displayed below the robot device. A photo of the specimen assembly is displayed in the bottom-right.

## 2.4. The force relaxation experiments

The same specimen assembly (i.e. the specimen, the cement and the aluminium cup) and the same single-leg stance loading configuration were used in both compression experiments.

One compression experiment was conducted in the micro-computed-tomography hutch of the Australian Synchrotron (Clayton, VIC, Australia) using a testing protocol described earlier [18]. In summary, the specimen was compressed via a spherically shaped polyethylene pressure socket of similar shape and stiffness to the natural acetabulum (figure 1). The pressure socket was displaced by a screw mechanism along the vertical axis of the potting cup. A low-friction x-y table minimized the transversal force components. The loading rate was $24 \pm 11$ N s$^{-1}$ (mean ± s.d.). The reaction force at the distal specimen was recorded continuously at 10 Hz for more than 30 min using a dedicated load cell (ME-measurement systems GmbH, Hennigsdorf, GE; capacity: 10 000 N and 500 Nm; error: 0.005%). The testing protocol provided a volume of images including the femoral head and the distal constraint using an isotropic pixel size equal to 30 µm. Imaging started a few minutes after loading of the specimen and lasted for 25.2 min, during which the specimen was under continuous X-ray exposure [18]. The images and the force profiles for the eight additional specimens reported here were obtained using an initial compression equal to one-fourth of the calculated fracture load.

The second compression experiment provided the force profiles under an initial compression below one-fifth of the estimated fracture load (figure 1) [25]. The specimens were instrumented using pre-wired stacked rosette strain gauges (KFG-3-120-D17-11L2M2S, Kyowa, Tokyo, Japan; gauge length: 3 mm) using a previously validated procedure for wet cadaveric bones [26,27]. The periosteum was completely removed. The bone surface was degreased using ethanol and a cocktail of acetone and 2-propanol. Bone pores were filled using two protective polyurethane layers (PU 140, HBM, Darmstadt, Germany) and then smoothened using fine sandpaper (#400). Ten strain gauges were placed at four anatomical aspects (medial, lateral, anterior and posterior) and three anatomical levels (femoral head, neck and metaphysis), avoiding regions with visible defects and high curvature. Strain gauges were attached to the bone surface using cyanoacrylate glue (ethyl 2-cyanoacrylate, Henkel, Rocky Hill, CT, USA), protected and waterproofed using three layers of polyurethane (PU 140, HBM, Darmstadt, Germany) (figure 1). The specimen assembly was mounted on the lower plate of a custom-made hexapod robot controlled by six servo-controlled ball screw-driven actuators connecting the robot's upper and lower plates [28]. A nylon pressure socket of a similar shape to that used in the imaging experiment (diameter: 47.4 mm) was mounted on the top plate. The generalized force vector acting on the upper plate was recorded using a six-axis load cell (MC3A-6-1000, AMTI, Watertown, MA, USA).

The robot's initial position was manually adjusted to align the pressure socket and the femoral head, ensuring no impingement occurred between the pressure socket and the greater trochanter. Contact between the pressure socket and the femoral head was established from the force response of the specimen. The actuators' length causing a reaction force oriented at 8° in the frontal plane and below one-fifth of the calculated fracture load in magnitude was determined using an adaptive velocity-based load control algorithm [29]. The time to reach equilibrium was less than 5 s resulting in a loading rate equal to $111 \pm 50$ N s$^{-1}$. The robot position was held constant for 30 min. The reaction force was recorded at 1 Hz. Cortical strains were recorded using a modular data logger (cDAQ-9178, National Instrument Corporation, Austin, TX, USA) equipped with eight four-channel bridge analogic modules (NI 9237, National Instrument Corporation, Austin, TX, USA). A 0.5 V excitation of the strain gauge grid was used to prevent heating of the bone surface. Strains were recorded at 500 Hz. A trigger was used to synchronize the strain gauges, the robot's leg length and the load cell signal. At all times, specimens were wrapped in fabric tissue soaked with phosphate-buffered saline solution to maintain bone moisture.

## 2.5. The force relaxation model

The data for the eight additional specimens reported in the present study were pooled with the data for the four specimens reported earlier [18,20]. The stretched exponential function by Sasaki and co-workers [9] was modified by replacing the elastic modulus $E$ with the reaction force $F$, hence taking the form:

$$F(t) = A \times e^{-(t/\tau)\beta}; t = [0,30], \tag{2.1}$$

where $A$ is the scaling factor, $\beta$ is the shape factor and $\tau$ is the characteristic time. The characteristic time $\tau$, the shape factor $\beta$ and the scaling factor $A$ were calculated by fitting the relaxation function (equation (2.1)) to each of the 30 force recordings using nonlinear least squares (Matlab, The MathWorks Inc., Natick, USA). The convergence tolerance was set to $1 \times 10^{-6}$. The force at 5 and 30 min relaxation (F5 and F30) were obtained. The force ratio ($F_0/F_L$) was the ratio between peak force ($F_0$) and the estimated fracture load ($F_L$); the force step increment ($F_{SI}$) was calculated as the force increment between consecutive load steps; and the relative force decline after 30 min ($F_{FL}$) was calculated as the ratio between the force step increment ($F_{SI}$) and the amount of force decrement at 30 min relaxation (figure 1).

## 2.6. Data analysis

The effect of the non-bone parts of the specimen assembly and the two different devices on the relaxation of the bone was assessed by monitoring the deformation of the bone, ideally constant throughout the experiment, using the cortical strain measurements and the images. Cortical strains were preliminarily screened for outliers and six-order low-pass Butterworth filtered (50 Hz). Changes of the equivalent cortical strain were calculated for the entire duration of the 30 min relaxation experiment. The presence of movement artefacts in the images was monitored, visually, to ascertain that every movement that occurred during imaging (25.2 min) was comparable, or below, the pixel size (0.03 mm). The force profiles obtained using the two testing protocols and a variable number of imaging cycles were analysed using robust linear regression, and the residuals were compared using a two-sample $t$-test.

Age, height, weight, BMI and bone quality indicators (hip bone mass, BMD, area, osteoporosis and fracture load) were analysed using descriptive statistics.

The goodness of fit of the stretched exponential function was assessed by calculating the coefficient of determination and by comparing the initial force $F_0$ and the scaling factor $A$. The relationship between the different force parameters analysed, characteristic time, shape factor, age, body weight, BMC and density were studied using robust linear regression analysis. Since bone damage increases as the load increases [29], the effect of damage on force relaxation was studied by comparing the force profiles recorded using an initial compression above and below 40% of the estimated fracture loads. Force relaxation data are accessible on Figshare [30].

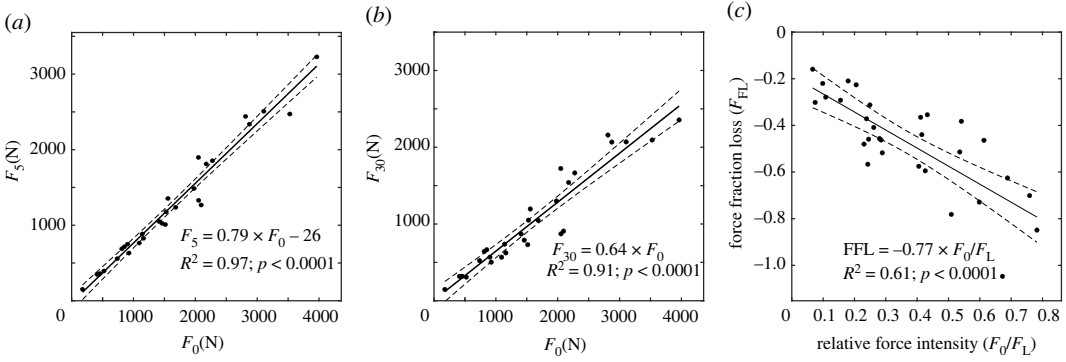

**Figure 2.** Linear regression analysis between the force $F_0$ and the force at 5- and 30 min relaxation (($a$) and ($b$)). On the right-hand side, the linear regression between the force ratio ($F_0/F_L$) and the relative force decline at 30 min ($F_{FL}$).

## 3. Results

Overall, 29 relaxation time histories were analysed. One relaxation time history was discarded because of data loss while recording. The images showed no substantial motion artefacts (electronic supplementary material, figure S4). The cortical strain changed by 1% on average during the first-minute relaxation (electronic supplementary material, figures S2, S3 and S4). By pooling the 29 measurements, the compressive force decreased by an average 21% ($R^2 = 0.97$, $p < 0.01$) after 5 min and by 36% ($R^2 = 0.91$, $p < 0.01$) after 30 min (figure 2). There was no statistical difference between the residuals between the two experimental protocols ($p > 0.45$). The force ratio ($F_0/F_L$) ranged between 0.07 and 0.78. The force fraction lost at 30 min, $F_{FL}$, was positively associated with the force ratio ($R^2 = 0.62$, $p < 0.01$), showing an average 7.7% increase in response to a 10% increase of the force ratio (figure 2).

The total hip bone mass, area, BMD and T-score were, respectively, $1.6 \pm 0.5$ g, $3.1 \pm 0.5$ cm$^2$, $0.5 \pm 0.1$ g cm$^{-2}$ and $-2.5 \pm 1.4$. The donors' height, weight, age and BMI were, respectively, $159 \pm 11$ cm, $90 \pm 34.7$ kg, $74.8 \pm 8.9$ years and $35.6 \pm 13.6$. The body weight declined by 2.6 kg per year of ageing ($R^2 = 0.37$, $p = 0.02$) as BMD declined by 0.01 g cm$^{-2}$ ($R^2 = 0.27$, $p = 0.05$). The calculated fracture load was 4840 N on average (range: 1998–8636 N) and was related to age ($R^2 = 0.37$, $p = 0.02$), weight ($R^2 = 0.34$, $p = 0.05$), BMI ($R^2 = 0.27$, $p = 0.05$) and BMD ($R^2 = 0.65$, $p < 0.001$). The fracture load decreased by 1067 N as BMD decreased by 0.1 g cm$^{-2}$ over 10 years of ageing (table 1).

The stretched exponential relaxation function fitted well to all force profiles analysed ($R^2 = 0.99$) (table 2) and displayed a strong association between the scaling factor A and the initial force $F_0$ ($R^2 > 0.89$, $p < 0.0001$). The characteristic time $\tau$ was related to the relative force decline at 30 min ($R^2 = 0.78$, $p < 0.01$). For an initial force below 40% of the estimated fracture load, the association between the characteristic time $\tau$ and the relative force decline at 30 min improved ($R^2 = 0.91$, $p < 0.01$) (figure 3e,f). The shape factor displayed a positive association to the force ratio ($R^2 = 0.31$, $p = 0.01$) (figure 3d and table 3) and was bi-linearly associated with the scaling factor and the logarithm of the characteristic time ($R^2 = 0.85$, $p < 0.001$) (figure 4). No association between the shape factor and the force ratio was found for an initial force exceeding 40% of femoral strength ($0.4 < F_0/F_L < 0.78$) (figure 3c and table 4).

**Table 2.** The initial force, $F_0$, the force ratio $F_0/F_L$, the force at 5 and 30 min relaxation, $F_5$ and $F_{30}$, the stretched exponential relaxation function fitted to the data and the goodness of fit ($R^2$ and RMSE). The experiment conducted using the robot device is reported in italics. Abbreviations: $A$ = scaling factor. $\beta$ = shape factor. $\tau$ = characteristic time (minutes).

| spec. ID | $F_0$ | $F_0/F_L$ | $F_5$ | $F_{30}$ | $A$ | $\beta$ | $\tau$ | $R^2$ | RMSE |
|---|---|---|---|---|---|---|---|---|---|
| 1 | 522 | 0.26 | 394 | 309 | 576 | 0.29 | 150 | 0.999 | 0.005 |
| 2 | 904 | 0.24 | 749 | 568 | 1084 | 0.29 | 132 | 0.995 | 0.010 |
| 2 | 1557 | 0.41 | 1354 | 1196 | 1658 | 0.26 | 2293 | 0.997 | 0.003 |
| 2 | 2050 | 0.54 | 1897 | 1723 | 2125 | 0.34 | 2829 | 0.998 | 0.002 |
| 2 | 2874 | 0.76 | 2337 | 2067 | 3217 | 0.20 | 1728 | 0.994 | 0.005 |
| 3 | 1161 | 0.28 | 827 | 624 | 1428 | 0.24 | 62 | 0.997 | 0.008 |
| 4 | 1093 | 0.23 | 764 | 568 | 1438 | 0.20 | 43 | 0.997 | 0.009 |
| 5 | 2053 | 0.41 | 1330 | 872 | 2279 | 0.31 | 36 | 0.984 | 0.013 |
| 5[a] | 3106 | 0.61 | 2508 | 2070 | 3256 | 0.30 | 379 | 0.999 | 0.003 |
| 5[b] | 3963 | 0.78 | 3228 | N/A | N/A | N/A | N/A | N/A | N/A |
| 6[a] | 1515 | 0.29 | 1009 | 731 | 1827 | 0.25 | 42 | 0.997 | 0.008 |
| 6 | 2176 | 0.41 | 1811 | 1541 | 2299 | 0.29 | 681 | 1.000 | 0.002 |
| 6 | 2812 | 0.54 | 2440 | 2159 | 2939 | 0.29 | 1683 | 0.999 | 0.001 |
| 6 | 3527 | 0.67 | 2472 | 2094 | 5000 | 0.12 | 91 | 0.975 | 0.015 |
| 7 | 1459 | 0.25 | 1026 | 789 | 1912 | 0.20 | 53 | 0.992 | 0.012 |
| 8 | 2094 | 0.24 | 1268 | 908 | 3965 | 0.14 | 2 | 0.981 | 0.024 |
| 9 | 1142 | 0.43 | 884 | 737 | 1387 | 0.19 | 332 | 0.997 | 0.006 |
| 10 | 925 | 0.28 | 631 | 503 | 1035 | 0.22 | 120 | 0.999 | 0.004 |
| 10 | 1408 | 0.43 | 1054 | 870 | 1496 | 0.25 | 345 | 1.000 | 0.002 |
| 10 | 1683 | 0.51 | 1237 | 1047 | 2014 | 0.17 | 360 | 0.997 | 0.005 |
| 10 | 1978 | 0.60 | 1486 | 1300 | 2419 | 0.15 | 757 | 0.992 | 0.007 |
| 10 | 2275 | 0.69 | 1853 | 1665 | 2465 | 0.19 | 4275 | 0.997 | 0.003 |
| 11 | 1527 | 0.25 | 1177 | 1050 | 1830 | 0.14 | 1960 | 0.990 | 0.007 |
| *1* | *410* | *0.21* | *350* | *318* | *429* | *0.22* | *6700* | *0.982* | *0.006* |
| *3* | *441* | *0.11* | *349* | *318* | *461* | *0.15* | *28 009* | *0.980* | *0.006* |
| *8* | *852* | *0.10* | *715* | *665* | *917* | *0.17* | *23 315* | *0.989* | *0.004* |
| *4* | *737* | *0.16* | *557* | *522* | *791* | *0.12* | *35 439* | *0.983* | *0.006* |
| *9* | *176* | *0.07* | *152* | *148* | *192* | *0.13* | *44 8958* | *0.834* | *0.013* |
| *12* | *812* | *0.18* | *687* | *642* | *888* | *0.15* | *53 038* | *0.982* | *0.005* |
| *11* | *456* | *0.07* | *360* | *319* | *519* | *0.17* | *1718* | *0.985* | *0.008* |
| mean | 1590 | 0.37 | 1230 | 977 | 1788 | 0.21 | 21 225 | 0.987 | 0.007 |
| s.d. | 967 | 0.21 | 777 | 605 | 1126 | 0.06 | 83 235 | 0.030 | 0.005 |
| CV | 61% | 0.57 | 63% | 62% | 63% | 30% | 392% | 3% | 71% |
| min | 176 | 0.07 | 152 | 148 | 192 | 0.12 | 2 | 0.834 | 0.001 |
| max | 3963 | 0.78 | 3228 | 2159 | 5000 | 0.34 | 448 958 | 1.000 | 0.024 |

[a]$F_{30}$ was estimated using the model fitted to approximately only 10 min of data recording.

[b]Fitting the model to the approximately 5 min of available data recording produced a shape factor of more than three standard deviations from the mean value. The estimated force at 30 min relaxation, $F_{30}$, and the fitting results were discarded.

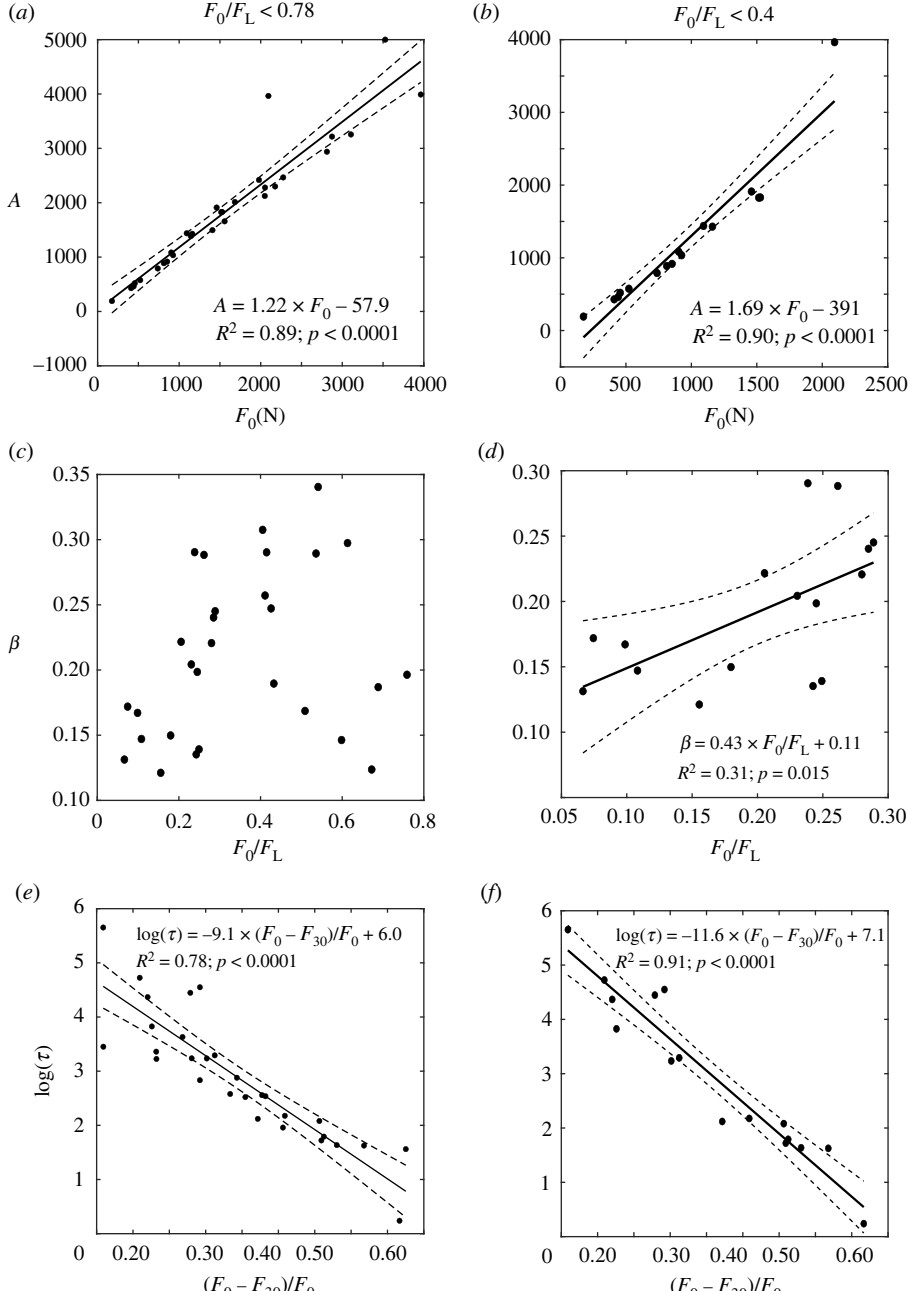

**Figure 3.** On the left-hand side, the robust linear regression analysis between $A$ and the initial force $F_0$ ($a$), $\beta$ and the force ratio $F_0/F_L$ ($c$), $\tau$ and the force fraction lost at 30 min relaxation $F_0 - F_{30}/F_0$ ($e$) executed by pooling the data. On the right-hand side (($b$), ($d$) and ($f$)), the same analysis is reported by pooling the data for a force ratio equal to or less than 0.4.

**Table 3.** The coefficient of determination calculated for the lower force range ($F_0/F_L$ less than 0.4). Only statistically significant values ($\alpha = 0.05$) are reported. The highest coefficient of determination per column is italicized.

| $R^2$ | $F_0$ | $F_0/F_L$ | $F_5$ | $F_0-F_5$ | $(F_0-F_5)/F_0$ | $F_{30}$ | $F_0-F_{30}$ | $(F_0-F_{30})/F_0$ | $F_{FL}$ | height | weight | age | BMI | bone mass | area | BMD | $A$ | $\beta$ | $\log(\tau)$ |
|---|---|---|---|---|---|---|---|---|---|---|---|---|---|---|---|---|---|---|---|
| $F_L$ | *0.36* | *0.37* | 0.27 | | | 0.42 | 0.22 | | | | *0.34* | 0.37 | 0.26 | *0.37* | | 0.72 | 0.37 | | |
| $F_0$ | | 0.36 | *0.96* | | *0.56* | 0.82 | 0.89 | 0.52 | 0.52 | | 0.20 | | | | | | 0.90 | | 0.51 |
| $F_0/F_LF_L$ | | | 0.35 | | 0.37 | 0.22 | 0.37 | 0.56 | 0.56 | | | | | | | | 0.22 | *0.31* | 0.57 |
| $F_5$ | | | | *0.73* | 0.38 | 0.94 | 0.73 | 0.38 | 0.38 | | 0.26 | | 0.24 | | | | 0.78 | | 0.40 |
| $F_0-F_5$ | | | | | | 0.53 | *0.98* | 0.66 | 0.66 | | | | | | | | *0.94* | | 0.60 |
| $(F_0-F_5)/F_0$ | | | | | | 0.20 | 0.87 | *0.87* | 0.87 | | | | | | | | 0.56 | | 0.68 |
| $F_{30}$ | | | | | | | 0.50 | | | | 0.30 | | 0.25 | | | 0.21 | 0.61 | | |
| $F_0-F_{30}$ | | | | | | | | 0.73 | 0.73 | | | | | | | | 0.92 | | 0.69 |
| $(F_0-F_{30})/F_0$ | | | | | | | | | *1.00* | | | | | | | | 0.50 | | *0.91* |
| $F_{FL}$ | | | | | | | | | | | | | | | | | 0.50 | | *0.91* |
| height | | | | | | | | | | | | | | | | | | | |
| weight | | | | | | | | | | | | *0.41* | *0.87* | | | | | | |
| age | | | | | | | | | | | | | 0.20 | 0.32 | | 0.37 | | | |
| BMI | | | | | | | | | | | | | | | | | | | |
| bone mass | | | | | | | | | | | | | | | *0.75* | | | | |
| area | | | | | | | | | | | | | | | | | | | |
| BMD | | | | | | | | | | | | | | | | | 0.24 | | |
| $A$ | | | | | | | | | | | | | | | | | | | 0.51 |
| $\beta$ | | | | | | | | | | | | | | | | | | | 0.20 |

**Table 4.** The coefficient of determination calculated by pooling the data. Only statistically significant values ($\alpha = 0.05$) are reported. The highest coefficient of determination per column is italicized.

| $R^2$ | $F_0$ | $F_0/F_L$ | $F_5$ | $F_0-F_5$ | $(F_0-F_5)/F_0$ | $F_{30}$ | $F_0-F_{30}$ | $(F_0-F_{30})/F_0$ | $F_{FL}$ | height | weight | age | BMI | bone mass | area | BMD | A | β | log($\tau$) |
|---|---|---|---|---|---|---|---|---|---|---|---|---|---|---|---|---|---|---|---|
| $F_L$ | *0.77* | | | | | | | | | | 0.14 | 0.31 | 0.13 | *0.47* | | 0.72 | | | |
| $F_0$ | | | *0.97* | *0.66* | | 0.91 | 0.76 | | 0.60 | | | 0.13 | | | | | | *0.89* | | |
| $F_0/F_L$ | | | 0.80 | 0.38 | | 0.80 | 0.45 | | 0.61 | | | | | | | | | 0.54 | | |
| $F_5$ | | | | 0.47 | | *0.97* | 0.60 | | 0.48 | | | 0.11 | | | | | | 0.76 | | |
| $F_0-F_5$ | | | | | *0.29* | 0.37 | *0.92* | 0.33 | 0.71 | | | | 0.14 | | | | | 0.82 | | 0.40 |
| $(F_0-F_5)/F_0$ | | | | | | 0.01 | 0.17 | *0.85* | 0.13 | | | | | | | | | | | 0.55 |
| $F_{30}$ | | | | | | | 0.46 | 0.30 | 0.42 | | | *0.16* | | | | | | 0.68 | | |
| $F_0-F_{30}$ | | | | | | | | | 0.67 | | | | 0.16 | | | | | 0.82 | | 0.44 |
| $(F_0-F_{30})/F_0$ | | | | | | | | | 0.18 | | | 0.16 | | | | | | | | *0.78* |
| $F_{FL}$ | | | | | | | | | | | | | | | | | | 0.61 | | 0.27 |
| height | | | | | | | | | | | | | *0.14* | | | | | | | |
| weight | | | | | | | | | | | | | *0.35* | *0.83* | | | | | | |
| age | | | | | | | | | | | | | | 0.14 | 0.22 | | 0.24 | | | |
| BMI | | | | | | | | | | | | | | | | | | | | |
| bone mass | | | | | | | | | | | | | | | | *0.17* | *0.82* | | | |
| area | | | | | | | | | | | | | | | | | | | | |
| BMD | | | | | | | | | | | | | | | | | | | | |
| A | | | | | | | | | | | | | | | | | | | | 0.22 |
| β | | | | | | | | | | | | | | | | | | | | |

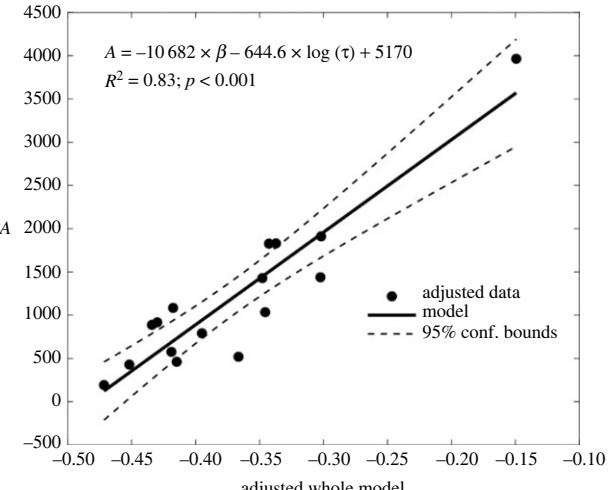

**Figure 4.** The bi-linear association between $A$, $\beta$ and log ($\tau$). By pooling the 29 force profiles, age was weakly associated with the force lost after 5 and 30 min ($R^2 = 0.14$–$0.16$) and the fracture load ($R^2 = 0.31$). The logarithm of the characteristic time, $\tau$, increased with age ($R^2 = 0.37$, $p = 0.03$) by pooling one force profile per specimen spanning a narrow range of the initial compression ($F_0/F_L = 0.23$–$0.43$; $n = 11$), explaining 35% of the variation of the logarithm of the characteristic time in the entire dataset. No association was found between age, scaling factor, $A$, shape factor, $\beta$, and characteristic time, $\tau$, either in the whole dataset or by excluding the force profiles with the highest initial compression ($F_0/F_L < 0.4$).

## 4. Discussion

The present study aimed at (i) testing the suitability of earlier bone stress relaxation theories for predicting the force relaxation response of the human femur and (ii) falsifying the hypothesis that ageing modulates the femur's relaxation response. The force relaxation of the femur was measured for 12 female donors whose age at death ranged from 56 to 91 years using two experimental protocols and an initial compression ranging from moderate to subcritical. The stretched exponential function developed earlier for isolated bone cores described well the force relaxation response of the femur, although the shape factor and the characteristic time differed from those reported earlier for isolated bone cores [9]. The initial compression determined most of the femur's force relaxation response exhibiting a larger effect on force relaxation than that of age (figure 4). Though, the early relaxation response to subcritical loads was variable, probably due to damage occurring early after load application.

The specimens described the known decline of bone mass and strength with age in the population at advanced age [31,32], and the initial force was strongly associated with the force decline after 5 and 30 min relaxation ($R^2 > 0.91$) pooling the data obtained using the two different experimental protocols and a variable number of imaging cycles. This provides confidence in the consistency of the data and the suitability of the 12 specimens for studying the effect of age on bone mechanics. Furthermore, the modest 1% change of bone deformation seen in the strain measurements, supported by the movement smaller than, or comparable to, the voxel size (0.03 mm) observed in the images of the entire bone volume, quantifies the deviation from the ideal relaxation condition of constant strain throughout the experiment [33]. Interestingly, the first-minute relaxation during which most of the deformation changes occurred is similar to the characteristic relaxation time of the cartilage [34], suggesting that the cartilage may have played a major role in determining the observed strain changes.

The stretched exponential function developed by Sasaki and co-workers [9] for bovine cortical bone cores well described the femur's force relaxation in elderly women ($R^2 = 0.99$). However, the femur's characteristic time $\tau$ was strongly related to the force fraction lost at 30 min ($R^2 = 0.78$) presenting similar values to those reported earlier for bovine bone cores only for the upper end of the force fraction lost after 30 min (figure 3e,f). Therefore, the characteristic time of isolated bone cores currently available may not well represent the femur's force relaxation. Furthermore, the femur's force relaxation response can be described as a function of the initial compression alone (figure 4) when the relative initial force is moderate ($F_0/F_L < 0.4$). However, the early force relaxation response was variable and unrelated to the initial compression for subcritical levels of the relative initial force (figure 3c), possibly due to damage onset and progression occurring during, or early after, application of subcritical compressive loads [20]. Age was moderately associated with the logarithm of the characteristic time ($R^2 = 0.37$) only when the analysis was restricted to a narrow range of the initial

compression ($F_0/F_L = 0.23$–$0.43$), explaining 35% of the variation of $\tau$ in the entire dataset. Therefore, the force relaxation response of the human femur is mostly determined by the amount of initial compression, relative to its strength.

The present work has limitations. First, the four specimens progressively loaded to fracture did not allow the specimen to recover between load steps (figure 1, B1), possibly contributing to the increased scatter of the data in the higher force range (figures 3 and 4). Nevertheless, the present study provides the compound effect of several mechanisms determining the force relaxation response in the human femur, including bone tissue distribution, dynamic properties, damage and poroelasticity. Increasing the number of specimens tested to fracture and constitutive models of bone viscoelasticity and damage may help to elucidate the separate contribution of the different underlying mechanisms of the femur's relaxation response [13–15]. Second, the two different testing protocols potentially increased the scatter of the data and affected the statistical analysis. Nevertheless, the strong association between the initial force and the relaxation response observed across all the data, and the analysis of the residuals, provides confidence in the generality (not methodology dependent) of the present conclusions. Third, the experiments were conducted at room temperature (23–26°C) leaving unresolved the effect of temperature on the force relaxation response. Fourth, the single time point and the pixel size in the images (i.e. 30 µm) did not allow to capture the relative movement of collagen molecules thought to cause the fast relaxation process of bone. Nevertheless, the present study confirms the suitability of the fast relaxation model developed earlier by Sasaki and co-workers [9] and provides the characteristic time and the shape factor for describing the force relaxation of human femurs. Finally, the 30 min relaxation experiment reported here is much shorter than the 4–8 h used in early relaxation experiments in excised bone cores [35]. Nevertheless, the relaxation response observed here describes most of the force relaxation response of the human femur.

In conclusion, the fast relaxation developed by Sasaki and co-workers [9] can be used to describe force relaxation of the human femur, although the characteristic time of femur's force relaxation response differs from that measured from bone cores and it is strongly related to the initial compression, relative to the femur's strength. The effect of age on force relaxation is modest compared with the effect of age on bone mass and strength, and the effect on femur relaxation of the initial compression. These results provide a quantitative assessment of force relaxation in human femora, expand current knowledge of bone dynamic properties and help to bridge the gap between relaxation experiments in isolated bone cores and entire bone organs.

Ethics. Ethics clearance (Project no. 6380) was obtained from the institutional Social and Behavioural Research Ethics Committee (SBREC).

Data accessibility. Force data are accessible from the supplementary Excel file and via Figshare: https://doi.org/10.6084/m9.figshare.c.5953405. The datasets contain 29 measurements of the 30 min relaxation response from 12 entire proximal femora specimens from female donors. The compressive force, one data point per second, applied to the specimen for more than 30 min using two different testing instrumentations is reported in two different sheets (sheet no. 1 and sheet no. 2). The header (ID no.) of each column corresponds to the row number, ascending order, in table 2. The grey shaded values were reconstructed using the model due to partial loss of the data ($R$-squared = 0.99).

Authors' contributions. S.M.: conceptualization, data curation, formal analysis, funding acquisition, investigation, methodology, project administration.

Conflict of interest declaration. I declare I have no competing interests.

Funding. Funding from the Australian Research Council (DP180103146; FT180100338; IC190100020) and the Australian Synchrotron (Clayton, VIC, Australia) are also gratefully acknowledged.

Acknowledgements. A special thanks to Dhara Amin for operating the hexapod robot, Egon Perilli, who contributed to the time-elapsed imaging experiment, and John Costi, who made available the BIL at Flinders University.

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
