## [Peer Review File · Royal Society Open Science]

Review History

RSOS-210479.R0 (Original submission)

Review form: Reviewer 1

Is the manuscript scientifically sound in its present form?

Yes

Are the interpretations and conclusions justified by the results?

Yes

Is the language acceptable?

Yes

Do you have any ethical concerns with this paper?

No

Have you any concerns about statistical analyses in this paper?

No

Recommendation?

Accept with minor revision (please list in comments)

Comments to the Author(s)

This paper represents an investigation of the relaxation behaviour of $n=12$ proximal femurs which have been loaded in multiple compression steps ($n=4$) or loaded in a single compression step ($n=8$). Two relaxation modes were identified, one below 40% of the femur failure load and one above this threshold. The manuscript has been transferred to RSOS following a previous review and editor decision and the author has done a very good job in answering comments by the previous reviewers.

The manuscript is very well written the aims of the study are clear and hypotheses are stated. The manuscript builds on two previous studies in *Acta Biomaterialia* and *JMBBM* [18, 19] which are excellent investigations. This generates, however, an impression of a smallest publishable unit approach as the presented material would have been a wonderful section in one of the previous studies. Methodologically, the current manuscript fits data to a stretched exponential function and performs (multi)linear regressions to identify potential relationships. However, there is no constitutive model used to aid the analyses so that the observations remain phenomenological.

There is certainly novelty in the performing relaxation tests on whole femurs and analysing the results with regards to donor age and BMD. However, the method is largely an extended repetition of previous experiments [18, 19]. The results are in line with the state-of-the-art and add an organ level component to this body of data. However, since the data is presented separated from the synchrotron radiation experiments [18, 19] under which the tests were actually performed it is not possible to gain a deeper insight into the viscous portions of bone's structure function relationship. This would have been a really exciting contribution.

A major concern is that femurs seem to have been tested under synchrotron radiation in previous experiments [18, 19]. While the structure of the organ may shadow potential radiation damage (see e.g. the brittle trabecular fractures in the samples of Thurner et al. (2006) that are suspiciously atypical in comparison to physiologic ones), several authors reported a detrimental effect of synchrotron radiation on material properties including in relaxation. There is, therefore, a risk that the results are influenced to an extent that does not allow to regard them as quasi-physiologic. Context for this has not been provided and a discussion of this impact is missing. Please note, if these femurs have not been tested under radiation, this comment should be disregarded.

Just as a side note: This paper is a direct continuation of [18, 19] and I am wondering whether other authors deserve credit here. Most of the methods and the actual experiments, for example, have been developed and performed as part of a previous experimental campaign as written by the author. I interpret good scientific practice so that credit is given to colleagues who have made a significant contribution.

Some minor comments below:

Line 21: It should be either "... the present study was that age modulated the force relaxation ..." or "... the present study is that age modulates the force relaxation ..."

Line 27: Just a suggestion based on taste, I'd remove the 'x' as the multiplication symbol I find this often misleading.

Lines 68-69: Damage is a well observable process in bone cores. It may well modulate relaxation behaviour if considered under your hypothesis.

Line 74: change “the best of the author” to “the best of the author’s”.

Line 81-82: There seems to be an error here: “from moderate to sb-critical”

Line 112: This is a relatively low density for femur tissue. How high is the error when calibrating density values outside the value range of the phantom?

Line 117: Do you refer to the element shape function when writing quadratic voxel mesh? This is misleading as voxels cannot be quadratic. Maybe describe it as: “converted into a voxel-based finite element mesh with quadratic elements” or similar.

Line 118: I would change “by Schileo at al. (2014) [22].” to match the citation style throughout the paper.

Line 126-161: There are a lot of references to previous studies which triggers the question what is novel here (see also my general comment above)?

Line 205: What do you mean by ‘pooling the data’? How was data pooled?

Lines 224-226: There is no damage metric defined here. In line with the suggestions by previous reviewers I suggest abstaining from that damage investigation. Also, the discussion clearly indicates that damage was not investigated (lines 314-315).

Line 255-256: Would you have expected otherwise? This type of function is extremely adaptable and this fit is no surprise at all. It is, however, not an expression of quality, as this function does not shed light on actual constitutive behaviour. It is a mere phenomenologic description of relaxation.

Line 257-258: A must be related to the peak force as it is an amplitude whereas the exponential represents a ‘damper’.

Line 265: The separation into groups below and above a force ratio of 0.4 is not clear. I am aware of the answer to previous reviewers but this rational should be made clear in the manuscript including the motivation and data for such a split

Figure 5: I am not sure what Fig 5 adds to the results at $t = 0$, the exponent becomes 1 and, thus, $A = F(0)$ according to eqn (1).

Line 287: change “white women donors” to “white female donors”

Review form: Reviewer 2

Is the manuscript scientifically sound in its present form?

No

Are the interpretations and conclusions justified by the results?

No

Is the language acceptable?

Yes

Do you have any ethical concerns with this paper?

Yes

Have you any concerns about statistical analyses in this paper?

No

Recommendation?

Reject

Comments to the Author(s)

GENERAL COMMENTS

This paper presents a re-analysis of previously published experiments. This causes some major limitations of this manuscript:

- A) The aim itself is not quite clear. What does the author ultimately aim to demonstrate?
- B) The experimental setup and protocol were clearly not designed to assess bone relaxation. Indeed, the force relaxation observed is the sum of several contributions:
- The bone, sure;
 - The cartilage, which is definitely viscoelastic;
 - The polyethylene cup used to deliver the force to the head;
 - The acrylic cement used to constrain the distal end (due to the long lever arm, a small deformation of the cement can result in a relatively large deflection of the head).

Due to the experimental design, it is not possible to ascertain the weight of the different components. For instance, if the cartilage relaxed some tenths of millimeter, this was probably sufficient to cause a drop of the force of some hundred N; as bone tissue is quite stiff, this is likely to result in a small variation of the strain measured by the strain gauges. Indeed, a steady trend of the strain variation is reported in Supplementary Fig. 2.

- C) Data come from two test campaigns:
- In one session, the bone was additionally exposed to large amount of X-ray radiation
 - Was the polyethylene cup used in the two campaigns identical?
 - Was the cement constraint in the two campaigns identical?

These are additional confounding factors that cannot be ignored, but it is not possible to ascertain their contribution.

For these reasons, I am afraid that the results and conclusions are not sufficiently robust to grant publication.

SPECIFIC COMMENTS

Two line-numbering series appear, one from the Author, the other one by the system. The comments below refer to the line numbers next to the text.

The paper is generally well-written and the images are clear. Here are some detailed comments:

1. INTRODUCTION, line 74: Did they observe “no change”, or the changes were “not statistically significant”?

2. INTRODUCTION, line 76-77: The present study does not really “test the validity of earlier theories”. What it does, is test their suitability to describe the behaviour of entire bone by extrapolating them from the tissue level.
3. INTRODUCTION, line 85-86: I would assume that the ethics clearance belongs to the M&M, not to the intro.
4. MATERIALS AND METHODS, lines 130-131: “constant deformation” can be deceiving as it is possible that some components relaxed significantly (see major comments above). What was constant was the position of the two ends. We don’t know exactly what happened between them.
5. MATERIALS AND METHODS, lines 135: “minimal deformation changes” is not a scientific description. This should be reported in quantitative terms.
6. MATERIALS AND METHODS, lines 145: “appreciable movement artefacts” is not a scientific description. This should be reported in quantitative terms. It is possible that movements were satisfactorily small for the original study, but possibly too large for the purposes of the present analysis.
7. MATERIALS AND METHODS, lines 219 (and in the entire paper, figures, tables): Is there a specific reason for calling “Fs” the failure force? It is not mnemonic, and can easily be misread as “F5” (which is also used in the paper).
8. RESULTS, line 242-243: you must provide an explanation of disregarding one specimen.
9. DISCUSSION and CONCLUSIONS: should be entirely revisited to account for the major limitations described above.

Decision letter (RSOS-210479.R0)

Dear Professor martelli

The Editors assigned to your paper RSOS-210479 "AGE MODULATES BMD AND STRENGTH BUT NOT FORCE RELAXATION IN HUMAN FEMORA" have made a decision based on their reading of the paper and any comments received from reviewers.

Regrettably, in view of the reports received, the manuscript has been rejected in its current form. However, a new manuscript may be submitted which takes into consideration these comments.

We invite you to respond to the comments supplied below and prepare a resubmission of your manuscript. Below the referees' and Editors' comments (where applicable) we provide additional requirements. We provide guidance below to help you prepare your revision.

Please note that resubmitting your manuscript does not guarantee eventual acceptance, and we do not generally allow multiple rounds of revision and resubmission, so we urge you to make every effort to fully address all of the comments at this stage. If deemed necessary by the Editors,

your manuscript will be sent back to one or more of the original reviewers for assessment. If the original reviewers are not available, we may invite new reviewers.

Please resubmit your revised manuscript and required files (see below) no later than 16-Jan-2022. Note: the ScholarOne system will 'lock' if resubmission is attempted on or after this deadline. If you do not think you will be able to meet this deadline, please contact the editorial office immediately.

Please note article processing charges apply to papers accepted for publication in Royal Society Open Science (<https://royalsocietypublishing.org/rsos/charges>). Charges will also apply to papers transferred to the journal from other Royal Society Publishing journals, as well as papers submitted as part of our collaboration with the Royal Society of Chemistry (<https://royalsocietypublishing.org/rsos/chemistry>). Fee waivers are available but must be requested when you submit your manuscript (<https://royalsocietypublishing.org/rsos/waivers>).

Thank you for submitting your manuscript to Royal Society Open Science and we look forward to receiving your resubmission. If you have any questions at all, please do not hesitate to get in touch.

on behalf of Dr Marco Palanca (Associate Editor) and Pietro Cicuta (Subject Editor)
openscience@royalsociety.org

Reviewer comments to Author:

Reviewer: 1

Comments to the Author(s)

This paper represents an investigation of the relaxation behaviour of $n=12$ proximal femurs which have been loaded in multiple compression steps ($n=4$) or loaded in a single compression step ($n=8$). Two relaxation modes were identified, one below 40% of the femur failure load and one above this threshold. The manuscript has been transferred to RSOS following a previous review and editor decision and the author has done a very good job in answering comments by the previous reviewers.

The manuscript is very well written the aims of the study are clear and hypotheses are stated. The manuscript builds on two previous studies in *Acta Biomaterialia* and *JMBBM* [18, 19] which are excellent investigations. This generates, however, an impression of a smallest publishable unit approach as the presented material would have been a wonderful section in one of the previous studies. Methodologically, the current manuscript fits data to a stretched exponential function and performs (multi)linear regressions to identify potential relationships. However, there is no constitutive model used to aid the analyses so that the observations remain phenomenological.

There is certainly novelty in the performing relaxation tests on whole femurs and analysing the results with regards to donor age and BMD. However, the method is largely an extended repetition of previous experiments [18, 19]. The results are in line with the state-of-the-art and add an organ level component to this body of data. However, since the data is presented separated from the synchrotron radiation experiments [18, 19] under which the tests were actually performed it is not possible to gain a deeper insight into the viscous portions of bone's structure function relationship. This would have been a really exciting contribution.

A major concern is that femurs seem to have been tested under synchrotron radiation in previous experiments [18, 19]. While the structure of the organ may shadow potential radiation damage (see e.g. the brittle trabecular fractures in the samples of Thurner et al. (2006) that are suspiciously atypical in comparison to physiologic ones), several authors reported a detrimental effect of synchrotron radiation on material properties including in relaxation. There is, therefore, a risk that the results are influenced to an extent that does not allow to regard them as quasi-physiologic. Context for this has not been provided and a discussion of this impact is missing. Please note, if these femurs have not been tested under radiation, this comment should be disregarded.

Just as a side note: This paper is a direct continuation of [18, 19] and I am wondering whether other authors deserve credit here. Most of the methods and the actual experiments, for example, have been developed and performed as part of a previous experimental campaign as written by the author. I interpret good scientific practice so that credit is given to colleagues who have made a significant contribution.

Some minor comments below:

Line 21: It should be either "... the present study was that age modulated the force relaxation ..." or "... the present study is that age modulates the force relaxation ..."

Line 27: Just a suggestion based on taste, I'd remove the 'x' as the multiplication symbol I find this often misleading.

Lines 68-69: Damage is a well observable process in bone cores. It may well modulate relaxation behaviour if considered under your hypothesis.

Line 74: change "the best of the author" to "the best of the author's".

Line 81-82: There seems to be an error here: "from moderate to sb-critical"

Line 112: This is a relatively low density for femur tissue. How high is the error when calibrating density values outside the value range of the phantom?

Line 117: Do you refer to the element shape function when writing quadratic voxel mesh? This is misleading as voxels cannot be quadratic. Maybe describe it as: "converted into a voxel-based finite element mesh with quadratic elements" or similar.

Line 118: I would change "by Schileo et al. (2014) [22]." to match the citation style throughout the paper.

Line 126-161: There are a lot of references to previous studies which triggers the question what is novel here (see also my general comment above)?

Line 205: What do you mean by 'pooling the data'? How was data pooled?

Lines 224-226: There is no damage metric defined here. In line with the suggestions by previous reviewers I suggest abstaining from that damage investigation. Also, the discussion clearly indicates that damage was not investigated (lines 314-315).

Line 255-256: Would you have expected otherwise? This type of function is extremely adaptable and this fit is no surprise at all. It is, however, not an expression of quality, as this function does

not shed light on actual constitutive behaviour. It is a mere phenomenologic description of relaxation.

Line 257-258: A must be related to the peak force as it is an amplitude whereas the exponential represents a 'damper'.

Line 265: The separation into groups below and above a force ratio of 0.4 is not clear. I am aware of the answer to previous reviewers but this rational should be made clear in the manuscript including the motivation and data for such a split

Figure 5: I am not sure what Fig 5 adds to the results at $t = 0$, the exponent becomes 1 and, thus, $A = F(0)$ according to eqn (1).

Line 287: change "white women donors" to "white female donors"

Reviewer: 2

Comments to the Author(s)

GENERAL COMMENTS

This paper presents a re-analysis of previously published experiments. This causes some major limitations of this manuscript:

A) The aim itself is not quite clear. What does the author ultimately aim to demonstrate?

B) The experimental setup and protocol were clearly not designed to assess bone relaxation.

Indeed, the force relaxation observed is the sum of several contributions:

- The bone, sure;
- The cartilage, which is definitely viscoelastic;
- The polyethylene cup used to the deliver the force to the head;
- The acrylic cement used to constrain the distal end (due to the long lever arm, a small deformation of the cement can result in a relatively large deflection of the head.

Due to the experimental design, it is not possible to ascertain the weight of the different components. For instance, if the cartilage relaxed some tenths of millimeter, this was probably sufficient to cause a drop of the force of some hundred N; as bone tissue is quite stiff, this is likely to result in a small variation of the strain measured by the strain gauges. Indeed, a steady trend of the strain variation is reported in Supplementary Fig. 2.

C) Data come from two test campaigns:

- In one session, the bone was additionally exposed to large amount of X-ray radiation
- Was the polyethylene cup used in the two campaigns identical?
- Was the cement constraint in the two campaigns identical?

These are additional confounding factors that cannot be ignored, but it is not possible to ascertain their contribution.

For these reasons, I am afraid that the results and conclusions are not sufficiently robust to grant publication.

SPECIFIC COMMENTS

Two line-numbering series appear, one from the Author, the other one by the system. The comments below refer to the line numbers next to the text.

The paper is generally well-written and the images are clear. Here are some detailed comments:

1. INTRODUCTION, line 74: Did they observe “no change”, or the changes were “not statistically significant”?
2. INTRODUCTION, line 76-77: The present study does not really “test the validity of earlier theories”. What it does, is test their suitability to describe the behaviour of entire bone by extrapolating them from the tissue level.
3. INTRODUCTION, line 85-86: I would assume that the ethics clearance belongs to the M&M, not to the intro.
4. MATERIALS AND METHODS, lines 130-131: “constant deformation” can be deceiving as it is possible that some components relaxed significantly (see major comments above). What was constant was the position of the two ends. We don’t know exactly what happened between them.
5. MATERIALS AND METHODS, lines 135: “minimal deformation changes” is not a scientific description. This should be reported in quantitative terms.
6. MATERIALS AND METHODS, lines 145: “appreciable movement artefacts” is not a scientific description. This should be reported in quantitative terms. It is possible that movements were satisfactorily small for the original study, but possibly too large for the purposes of the present analysis.
7. MATERIALS AND METHODS, lines 219 (and in the entire paper, figures, tables): Is there a specific reason for calling “Fs” the failure force? It is not mnemonic, and can easily be misread as “F5” (which is also used in the paper).
8. RESULTS, line 242-243: you must provide an explanation of disregarding one specimen.
9. DISCUSSION and CONCLUSIONS: should be entirely revisited to account for the major limitations described above.

===PREPARING YOUR MANUSCRIPT===

While not essential, it will speed up the preparation of your manuscript proof if accepted if you format your references/bibliography in Vancouver style (please see

<https://royalsociety.org/journals/authors/author-guidelines/#formatting>). You should include DOIs for as many of the references as possible.

===PREPARING YOUR REVISION IN SCHOLARONE===

Author's Response to Decision Letter for (RSOS-210479.R0)

See Appendix A.

RSOS-211567.R0

Review form: Reviewer 1

Is the manuscript scientifically sound in its present form?

Yes

Are the interpretations and conclusions justified by the results?

Yes

Is the language acceptable?

Yes

Do you have any ethical concerns with this paper?

No

Have you any concerns about statistical analyses in this paper?

No

Recommendation?

Accept as is

Comments to the Author(s)

The responded to all my questions and comments. There are a couple of points that I consider under scientific debate which must respect different opinion and for which I am grateful.

The sticky point is that I still believe that this is largely an extended repetition of previous experiments [18, 19] and, thus, an overall incremental manuscript. This has also been flagged up by reviewer 2 and my comments in the initial review with regards to novel aspects of the paper

do not serve to counter the comments of reviewer 2. This is, however, an editorial issue and not an issue of the overall content of the manuscript and the author's responses, which were written well.

Review form: Reviewer 2

Is the manuscript scientifically sound in its present form?

No

Are the interpretations and conclusions justified by the results?

No

Is the language acceptable?

Yes

Do you have any ethical concerns with this paper?

No

Have you any concerns about statistical analyses in this paper?

No

Recommendation?

Reject

Comments to the Author(s)

It is difficult to follow the author's responses as numbering and lettering of the points are mixed (eg. my first general comment, which I had labelled as "A" is listed just as "26", my former point "B" has become "27 B" etc).

Additionally, it seems that the authors have not considered nor responded to the specific comments on the previous version. Therefore, I am not going to waste more time writing more specific comments, I will just focus on the main ones

The Author should have checked more carefully, before submitting.

The Author stresses the different views of the two Reviewers. The fact that two Reviewers have different opinions is not uncommon (for instance, because they have different backgrounds), and does not undermine the relevance of the respective comments.

A) In the view of this Reviewer, the aim itself is not quite clear. It is true that two aims are stated. What is not clear is how this relates to the title (where strength is mentioned). This problem remains in the revised paper, where this problem is unsolved. Also, my specific comment #2 to the previous version related to the definition of the aims of this study. Moreover, for what concerns Question 2: a hypothesis can be falsified by a positive finding. In your case, your study only found a correlation that was not significant. This could be due to a true lack of correlation, but also to having tested a very small number of specimens, covering a narrow range of ages of post-menopausal women.

B) The answer to my concern about confounding factors is not convincing. It is obvious that "total displacement (deformation) is the sum of the displacement (deformation) of the different parts, not the force, which is equally applied to the pressure socket, the cartilage,

the bone, and the distal constraint, for equilibrium". Exactly for this reason: if any of the components (the bone and/or the cement and/or the socked and/or the cartilage relaxates, you will see a drop of the force.

C) Data come from two test campaigns. The experimental setup and protocol were clearly not designed to assess bone relaxation as one has to move through datasets and assumptions to compare data that are not comparable: Some specimens were tested multiple times (hence have a different weight in the stats), some underwent massive x-radiation, strain was measured, but only in a subset of specimens, etc. (even the reply about the cement is unclear: "was the cement identical?" "No, it was the same cement constraint mounted to both devices"... and then line 205 does not mention the cement

These are additional confounding factors that cannot be ignored, but it is not possible to ascertain their contribution.

D) Methods are presented in a very confusing way, for instance:

- To understand the actual force values delivered to the different specimens one has to gather this information from the text, plus the tables, plus reading another manuscript.
- Knowing the force value and the force ratio would help the reader understand, for instance, if the two experiments (the radiation-exposed one and the robot one) overlapped in terms of force ranges. This is fundamental, for instance, to interpret Fig. 4 (formerly fig. 3)
- What is the logical order of the rows in Table 2?

Decision letter (RSOS-211567.R0)

Dear Professor martelli

The Editors assigned to your paper RSOS-211567 "AGE MODULATES BMD AND STRENGTH BUT NOT FORCE RELAXATION IN HUMAN FEMORA" have made a decision based on their reading of the paper and any comments received from reviewers.

Regrettably, in view of the reports received, the manuscript has been rejected in its current form. However, a new manuscript may be submitted which takes into consideration these comments.

We invite you to respond to the comments supplied below and prepare a resubmission of your manuscript. Below the referees' and Editors' comments (where applicable) we provide additional requirements. We provide guidance below to help you prepare your revision.

Please note that resubmitting your manuscript does not guarantee eventual acceptance, and we do not generally allow multiple rounds of revision and resubmission, so we urge you to make every effort to fully address all of the comments at this stage. If deemed necessary by the Editors, your manuscript will be sent back to one or more of the original reviewers for assessment. If the original reviewers are not available, we may invite new reviewers.

Please resubmit your revised manuscript and required files (see below) no later than 03-Aug-2022. Note: the ScholarOne system will 'lock' if resubmission is attempted on or after this deadline. If you do not think you will be able to meet this deadline, please contact the editorial office immediately.

Please note article processing charges apply to papers accepted for publication in Royal Society Open Science (<https://royalsocietypublishing.org/rsos/charges>). Charges will also apply to papers transferred to the journal from other Royal Society Publishing journals, as well as papers submitted as part of our collaboration with the Royal Society of Chemistry (<https://royalsocietypublishing.org/rsos/chemistry>). Fee waivers are available but must be requested when you submit your manuscript (<https://royalsocietypublishing.org/rsos/waivers>).

Thank you for submitting your manuscript to Royal Society Open Science and we look forward to receiving your resubmission. If you have any questions at all, please do not hesitate to get in touch.

on behalf of Dr Marco Palanca (Associate Editor) and Pietro Cicuta (Subject Editor)
openscience@royalsociety.org

Editor comments:

The reviews were not entirely negative, and perhaps the manuscript can be improved and resubmitted, but the reviewers did not think their concerns were met fully, and this clearly requires extra work.

Reviewer comments to Author:

Reviewer: 2

Comments to the Author(s)

It is difficult to follow the author's responses as numbering and lettering of the points are mixed (eg. my first general comment, which I had labelled as "A" is listed just as "26", my former point "B" has become "27 B" etc).

Additionally, it seems that the authors have not considered nor responded to the specific comments on the previous version. Therefore, I am not going to waste more time writing more specific comments, I will just focus on the main ones

The Author should have checked more carefully, before submitting.

The Author stresses the different views of the two Reviewers. The fact that two Reviewers have different opinions is not uncommon (for instance, because they have different backgrounds), and does not undermine the relevance of the respective comments.

A) In the view of this Reviewer, the aim itself is not quite clear. It is true that two aims are stated. What is not clear is how this relates to the title (where strength is mentioned). This problem remains in the revised paper, where this problem is unsolved. Also, my specific comment #2 to the previous version related to the definition of the aims of this study. Moreover, for what concerns Question 2: a hypothesis can be falsified by a positive finding. In your case, your study only found a correlation that was not significant. This could be due to a true lack of correlation, but also to having tested a very small number of specimens, covering a narrow range of ages of post-menopausal women.

B) The answer to my concern about confounding factors is not convincing. It is obvious that “total displacement (deformation) is the sum of the displacement (deformation) of the different parts, not the force, which is equally applied to the pressure socket, the cartilage, the bone, and the distal constraint, for equilibrium”. Exactly for this reason: if any of the components (the bone and/or the cement and/or the socket and/or the cartilage relaxates, you will see a drop of the force.

C) Data come from two test campaigns. The experimental setup and protocol were clearly not designed to assess bone relaxation as one has to move through datasets and assumptions to compare data that are not comparable: Some specimens were tested multiple times (hence have a different weight in the stats), some underwent massive x-radiation, strain was measured, but only in a subset of specimens, etc. (even the reply about the cement is unclear: “was the cement identical?” “No, it was the same cement constraint mounted to both devices”... and then line 205 does not mention the cement

These are additional confounding factors that cannot be ignored, but it is not possible to ascertain their contribution.

D) Methods are presented in a very confusing way, for instance:

- To understand the actual force values delivered to the different specimens one has to gather this information from the text, plus the tables, plus reading another manuscript.
- Knowing the force value and the force ratio would help the reader understand, for instance, if the two experiments (the radiation-exposed one and the robot one) overlapped in terms of force ranges. This is fundamental, for instance, to interpret Fig. 4 (formerly fig. 3)
- What is the logical order of the rows in Table 2?

Reviewer: 1

Comments to the Author(s)

The responded to all my questions and comments. There are a couple of points that I consider under scientific debate which must respect different opinion and for which I am grateful.

The sticky point is that I still believe that this is largely an extended repetition of previous experiments [18, 19] and, thus, an overall incremental manuscript. This has also been flagged up by reviewer 2 and my comments in the initial review with regards to novel aspects of the paper do not serve to counter the comments of reviewer 2. This is, however, an editorial issue and not an issue of the overall content of the manuscript and the author's responses, which were written well.

===PREPARING YOUR MANUSCRIPT===

Please ensure that you include an acknowledgements' section before your reference list/bibliography. This should acknowledge anyone who assisted with your work, but does not

qualify as an author per the guidelines at <https://royalsociety.org/journals/ethics-policies/openness/>.

If you have been asked to revise the written English in your submission as a condition of publication, you must do so, and you are expected to provide evidence that you have received language editing support. The journal would prefer that you use a professional language editing service and provide a certificate of editing, but a signed letter from a colleague who is a fluent speaker of English is acceptable. Note the journal has arranged a number of discounts for authors using professional language editing services (<https://royalsociety.org/journals/authors/benefits/language-editing/>).

===PREPARING YOUR REVISION IN SCHOLARONE===

- Ensure that your data access statement meets the requirements at <https://royalsociety.org/journals/authors/author-guidelines/#data>. You should ensure that you cite the dataset in your reference list. If you have deposited data etc in the Dryad repository, please include both the 'For publication' link and 'For review' link at this stage.
- If you are requesting an article processing charge waiver, you must select the relevant waiver option (if requesting a discretionary waiver, the form should have been uploaded at Step 3 'File upload' above).
- If you have uploaded ESM files, please ensure you follow the guidance at <https://royalsociety.org/journals/authors/author-guidelines/#supplementary-material> to include a suitable title and informative caption. An example of appropriate titling and captioning may be found at https://figshare.com/articles/Table_S2_from_Is_there_a_trade-off_between_peak_performance_and_performance_breadth_across_temperatures_for_aerobic_scope_in_teleost_fishes_/3843624.

Author's Response to Decision Letter for (RSOS-211567.R0)

See Appendix B.

Decision letter (RSOS-220301.R0)

Dear Professor Martelli,

I am pleased to inform you that your manuscript entitled "AGE MODULATES BMD AND STRENGTH BUT NOT FORCE RELAXATION IN HUMAN FEMORA" is now accepted for publication in Royal Society Open Science.

The proof of your paper will be available for review using the Royal Society online proofing system and you will receive details of how to access this in the near future from our production office (openscience_proofs@royalsociety.org). We aim to maintain rapid times to publication after

acceptance of your manuscript and we would ask you to please contact both the production office and editorial office if you are likely to be away from e-mail contact to minimise delays to publication. If you are going to be away, please nominate a co-author (if available) to manage the proofing process, and ensure they are copied into your email to the journal.

on behalf of Dr Marco Palanca (Associate Editor) and Pietro Cicuta (Subject Editor)
openscience@royalsociety.org

Associate Editor Comments to Author (Dr Marco Palanca):

The author addressed all points raised by the reviewers in the previous revision. The author clarified the methodologies and the findings of this work, and better described the relevance of the work. I'm confident that, in this version, the manuscript is sharp enough for being an interesting point of discussion for the scientific community.

Appendix A

RESPONSE TO THE REVIEWER COMMENTS

AGE MODULATES BMD AND STRENGTH BUT NOT FORCE RELAXATION IN HUMAN FEMORA

Saulo Martelli

Both reviewers provided several constructive comments. Reviewer #1 acknowledges the novel contribution of the manuscript, commends on the clarity of aim and hypothesis and he/she is concerned about the validity of the two different devices used to collect the data. In contrast, reviewer #2 considers the aim and hypothesis unclear and, similarly to reviewer #1, he/she is concerned about the validity of the experiment and particularly about the effect of cartilage and other non-bone components in the specimen assembly on the recorded force relaxation response. The manuscript has been substantially revised to provide quantitative information about the validity of the data used for the analysis and provide point-by-point answers to both reviewers. The reviewers' comments are reported in italics and the author's response is provided immediately below in plain text. Comments and answers are numbered continuously throughout the present response. A version of the revised manuscript with tracked changes immediately follow a clean version of the revised manuscript.

1. *Reviewer: 1*

This paper represents an investigation of the relaxation behaviour of n=12 proximal femurs which have been loaded in multiple compression steps (n=4) or loaded in a single compression step (n=8). Two relaxation modes were identified, one below 40% of the femur failure load and one above this threshold. The manuscript has been transferred to RSOS following a previous review and editor decision and the author has done a very good job in answering comments by the previous reviewers. The manuscript is very well written the aims of the study are clear and hypotheses are stated.

Thank you.

2. *The manuscript builds on two previous studies in Acta Biomaterialia and JMBBM [18, 19] which are excellent investigations. This generates, however, an impression of a smallest publishable unit approach as the presented material would have been a wonderful section in one of the previous studies.*

The problem of a virtuous approach to scientific publication is indeed a very interesting point. The present analysis was motivated by the opportunity, realized during the two studies of reference,

rather than planned a-priori by a systematic approach to publication. In the author's opinion, the different aim between the two earlier studies (i.e., description of the testing protocol and the fracture mechanism) and the present analysis (i.e., force relaxation) makes the benefit of merging the studies questionable. The contribution of the present study is clearly stated, as acknowledged by the reviewer, in the hypothesis and aim of the study (line 33):

“The aim of the present study was 1) to test the validity of earlier bone stress relaxation theories developed for bone cores for predicting the force relaxation response of human femora and 2) to falsify the hypothesis that aging modulates the femur’s relaxation response.”

3. Methodologically, the current manuscript fits data to a stretched exponential function and performs (multi)linear regressions to identify potential relationships. However, there is no constitutive model used to aid the analyses so that the observations remain phenomenological.

The reviewer is correct. This is clearly stated throughout the text (lines 38, 60, and 202). However, it should be noted that that the stretched exponential function used here was associated with the motion of the collagen molecules by Sasaki and co-workers (Goto et al., 1999; Nakayama et al., 1993) as opposed, for example, to the exponential decay function fitting well to the long-term relaxation response (> 70 hours), which has been associated with the sliding between cement lines. While certainly a phenomenological observation, this differentiates the present phenomenological model from common purely phenomenological studies where a base function completely unrelated to the underlying mechanism is fitted to the data. This point has been clarified further in the text. The text states (line 13):

“The fast process, thought to be governed by the relative motion of collagen molecules [7,8], explains the large majority of the stress decline over time and can be described by a stretched exponential decay function with a characteristic time equal to 25 – 45 minutes [9]. The slow process, thought to be governed by a viscous-like motion between cement lines [7,8], can be described by an exponential decay function with a characteristic time equal to 70 – 300 hours [9]”

line 170:

“The force relaxation profiles obtained in the present study were pooled with the images and force profiles obtained earlier [18,19]. The stretched exponential function by Sasaki and co-workers [9] was modified by replacing the elastic modulus E with the reaction force F , hence taking the form:”

Line 303:

“More femoral specimens tested to fracture and constitutive models of bone viscoelasticity and damage may help to elucidate the separate contribution of the different underlying mechanisms of the femur’s relaxation response [13–15]. The present study demonstrates that the force relaxation response of the human femur is mostly determined by the amount of force applied, relative to its strength, while the effect of age was too small to be detected.”

4. *There is certainly novelty in the performing relaxation tests on whole femurs and analysing the results with regards to donor age and BMD.*

Thank you.

5. *However, the method is largely an extended repetition of previous experiments [18, 19].*

The reviewer is correct. The aim of the study was, in fact, not that of developing a new method but rather that of obtaining a dataset larger enough for the statistical analysis, which is novel and original. This aspect has now been clarified in the text that now states (line 36):

“The femur’s force relaxation response was obtained from twelve elderly female donors between 56 and 91 years of age. Thirty force relaxation profiles spanning an initial compression from moderate to sub-maximal were obtained by pooling published data and novel measurements obtained with two established testing protocols. The stretched exponential relaxation function by Sasaki and co-workers [9] was fitted to the data. The relationship between force relaxation, age, body weight, bone mineral content, and density was studied using robust linear regression analysis.”

A paragraph has also been added at the beginning of the methods section to clarify the different sources of the data. The paragraph states (line 45):

“Ethics clearance (Project # 6380) was obtained from the institutional Social and Behavioural Research Ethics Committee (SBREC). The force relaxation profiles were obtained using two different custom-made compression devices. One device provided the force relaxation profile and concomitant microstructural computed-tomography images obtained at the Australian Synchrotron (Clayton VIC, Australia). The full description of the testing protocol and the force profiles for four specimens loaded to fracture were reported earlier [18,19]. In the present study, the force relaxation profiles for additional eight specimens subjected to an initial compression equal to one-fourth of the estimated fracture load were added to the analysis. The second device was a custom-made robot used to record the force relaxation profiles from the same eight specimens under an initial compression below one-fifth of the estimated fracture load. The entire database comprised 30 force relaxation profiles. The effect of the two different devices and a variable number (0 – 6) of imaging cycles on the force

measurement was assessed by comparing the force profiles. Also, the deviation from bone strain uniformity over time, attributable to the non-bone components of the specimen assembly and the performance of the testing device, was quantified using cortical strain gages and by visual inspection of the microstructural images (pixel size: 30 microns). Bone quality indicators were analyzed using descriptive statistics. The stretched exponential relaxation function by Sasaki and co-workers [9] was fitted to the data. The relationship between force relaxation, age, body weight, bone mineral content, and density was studied using robust linear regression analysis.”

6. *The results are in line with the state-of-the-art and add an organ level component to this body of data.*

Thank you.

7. *However, since the data is presented separated from the synchrotron radiation experiments [18, 19] under which the tests were actually performed it is not possible to gain a deeper insight into the viscous portions of bone’s structure function relationship. This would have been a really exciting contribution.*

It is unclear which viscous portion of the bone’s structure-function relationship the reviewer is referring to. The microstructural images (30 microns pixel size) provide a single time point and do not provide information about the motion of the collagen molecules, which would require a much smaller pixel size. This point is now clarified further in the text (line 319):

“Fourthly, the single time point and the pixel size in the images (i.e., 30 microns) did not allow to capture the relative movement of collagen molecules thought to cause the fast relaxation process of bone. Nevertheless, the present study confirms the validity of the fast relaxation model developed earlier by Sasaki and co-workers [9] and provides the characteristic time and the shape factor for describing the force relaxation of human femurs.”

8. *A major concern is that femurs seem to have been tested under synchrotron radiation in previous experiments [18, 19]. While the structure of the organ may shadow potential radiation damage (see e.g. the brittle trabecular fractures in the samples of Thurner et al. (2006) that are suspiciously atypical in comparison to physiologic ones), several authors reported a detrimental effect of synchrotron radiation on material properties including in relaxation. There is, therefore, a risk that the results are influenced to an extent that does not allow to regard them as quasi-physiologic. Context for this has not been provided and a discussion of this impact is missing. Please*

note, if these femurs have not been tested under radiation, this comment should be disregarded.

This is a very good point and the main concern during [19]. However, the present study cannot be directly compared to *Thurner et al. (2006)* because of the very different specimen size, the 3 mm thick aluminum wall, and the thick layer of tissue soaked with the saline solution used only in the present study. This likely determined a very different radiation per unit of bone volume, making the comparison between the two studies difficult, if not impossible. The protocol used here was developed by first filling the compressive stage with saline solution, letting no light through to the detector (black image), and then reducing the amount of saline solution, using a soaked fabric wrapped around the specimen, to enable light to reach the detector and sufficient contrast in the images. The ultimate deformation to fracture, after 5 – 7 imaging cycles, reached 15% [19] showing a large deformation unlike the brittle response observed by *Thurner et al. (2006)*. Furthermore, by pooling the data obtained using a variable number (1 to 7) of imaging cycles and the two different devices, the initial force F_0 alone explained more than 90% of the variation of the force recorded after 5- and 30-minute relaxation. This indicates that irradiation of the specimen had a moderate effect (< 9%) on the force relaxation response. It appears, therefore, that the experiment described here is more likely comparable to previous HR-QCT studies of entire bones under load, like *Jackman et al. (2016)*, rather than to studies of bone cores using synchrotron light like *Thurner et al. (2006)*. This point has been clarified further in the discussion. Nevertheless, the validity of the data used in the present study was a major concern of both reviewers and the manuscript has been substantially amended to provide a quantification of the effect of the different testing protocols, the viscoelastic effect of non-bone components, and the performance of the different devices used here. The text now states (line 58):

“Also, the deviation from bone strain uniformity over time, attributable to the non-bone components of the specimen assembly and the performance of the testing device, was quantified using cortical strain gages and by visual inspection of the microstructural images (pixel size: 30 microns). Bone quality indicators were analyzed using descriptive statistics.”

Line 186

“The consistency of the force relaxation measurements obtained using the two different testing devices and a variable number of imaging cycles was analyzed using robust linear regression. The change over time of bone deformation attributable to the non-bone components of the specimen assembly and the performance of the testing devices was assessed using both the cortical strain measurement and the images. The cortical strain measurements were preliminarily screened for

outliers and six-order low-pass Butterworth filtered (50 Hz). Changes of the equivalent cortical strain were calculated for the entire duration of the 30-minute relaxation experiment. A visual inspection of the images was conducted to ascertain no substantial movement, relative to the initial displacement, occurred during imaging (25.2 minutes) over the entire bone volume.”

And line 279:

“The strong correlation between the initial force and the force decline after 5 and 30 minutes ($R^2 > 0.91$) across all the force recordings obtained with different devices and variable imaging cycles provided confidence in the consistency of the data. Furthermore, the modest 1 % change of cortical strain during the first-minute relaxation is consistent with no substantial motion visible in the images across the entire bone volume and interestingly comparable to the characteristic relaxation time of the cartilage [30]. These modest bone strain changes are expected to cause similar changes in the force response of the bone specimen due to the linearity of the bone’s force-displacement relationship [31], hence indicating that most of the force relaxation reported here can be attributed to the bone while the cartilage may have played a role in the initial small drift of bone strain.”

9. *Just as a side note: This paper is a direct continuation of [18, 19] and I am wondering whether other authors deserve credit here. Most of the methods and the actual experiments, for example, have been developed and performed as part of a previous experimental campaign as written by the author. I interpret good scientific practice so that credit is given to colleagues who have made a significant contribution.*

The reviewer raises an important ethical issue. According to the Australia Code for the Responsible Conduct of Research, attributed authors should have made a significant contribution to two of the following aspects: 1) procurement of research data, where the acquisition involves significant intellectual judgment, planning, design, or input; 2) contribution of knowledge; 3) evaluation or interpretation of research data; 4) generation of significant sections of the research output or critical revisions contributing to its interpretation. The author has integrally designed, processed, and written the manuscript. Dhara Amin operated the hexapod robot. Egon Perilli contributed to setting up the time-elapsd imaging experiment during [19]. John Costi made available the BIL laboratory at Flinders University. These contributions and funding are gratefully acknowledged in the appropriate section of the main text (line 341):

“A special thanks to Dhara Amin for operating the hexapod robot, Egon Perilli, who contributed to the time-elapsd imaging experiment, and John Costi, who made available the BIL at Flinders University. Funding from the Australian Research Council (DP180103146; FT180100338;

IC190100020) and the Australian Synchrotron (Clayton, VIC, Australia) are also gratefully acknowledged.”

Some minor comments below:

10. Line 21: It should be either “... the present study was that age modulated the force relaxation ...” or “... the present study is that age modulates the force relaxation ...”

Thank you for this comment. The text has been amended accordingly.

11. Line 27: Just a suggestion based on taste, I'd remove the 'x' as the multiplication symbol I find this often misleading.

Done

12. Lines 68-69: Damage is a well observable process in bone cores. It may well modulate relaxation behaviour if considered under your hypothesis.

Precisely. Damage accumulates as load increases so its effect on relaxation can be observed by comparing the reaction force obtained using a low and high initial compression. The 40% of the estimated fracture load threshold was defined by observing that little bone exceeds yield below this level as opposed to strain levels increasing to 10 – 15% under sub-critical loads [19]. This is now clarified in the text, which now states (line 202):

“Since bone damage increases as the initial compression increases [29], the effect of damage on force relaxation were studied by comparing the force profiles recorded using an initial compression above and below 40% of the estimated fracture loads.”

13. Line 74: change “the best of the author” to “the best of the author’s”.

Thank you for this comment. The text has been amended accordingly.

14. Line 81-82: There seems to be an error here: “from moderate to sb-critical”

Corrected. Thank you.

15. Line 112: This is a relatively low density for femur tissue. How high is the error when calibrating density values outside the value range of the phantom?

We thank the reviewer for this comment. The range reported in the text did not include the densest object in the phantom. This error has now been corrected and the densest rod in the phantom (375.8 mg·cm⁻³) has been added to the text, which now states (line 81):

“. A phantom (Mindways Software, Inc., Austin, USA) with five samples of known dipotassium hydrogen phosphate density (K₂HPO₄ equivalent density range: 51.8 – 375.8 mg·cm⁻³) was scanned with the samples.”

16. Line 117: Do you refer to the element shape function when writing quadratic voxel mesh? This is misleading as voxels cannot be quadratic. Maybe describe it as: “converted into a voxel-based finite element mesh with quadratic elements” or similar.

Thank you for this comment. The text has been amended and now states (line 89):

“The voxels in the volume of images were converted into a mesh of quadratic hexahedral elements.”.

17. Line 118: I would change “by Schileo et al. (2014) [22].” to match the citation style throughout the paper.

Done. Thank you.

18. Line 126-161: There are a lot of references to previous studies which triggers the question what is novel here (see also my general comment above)?

This aspect has been clarified by adding a dedicated paragraph at the beginning of the methods section (line 45). The paragraph is copied and pasted here above with the author’s answer to comment 5.

19. Line 205: What do you mean by ‘pooling the data’? How was data pooled?

Data pooling is a process where data sets coming from different sources are combined. The text has been edited to improve clarity and it now states (line 170):

“The force relaxation profiles obtained in the present study were pooled with the images and force profiles obtained earlier [18,19].”

20. Lines 224-226: There is no damage metric defined here. In line with the suggestions by previous reviewers I suggest abstaining from that damage investigation. Also, the discussion clearly indicates that damage was not investigated (lines 314-315).

The damage process was, in fact, not investigated. However, comparing the relaxation response at low and sub-maximal loads, which cause minimal and sub-maximal levels of damage, can provide quantitative information about the effect of damage on the force relaxation response. Please refer to the author’s answer to comment 12 for details about how this point has been clarified in the text.

21. Line 255-256: *Would you have expected otherwise? This type of function is extremely adaptable, and this fit is no surprise at all. It is, however, not an expression of quality, as this function does not shed light on actual constitutive behaviour. It is a mere phenomenologic description of relaxation.*

Sasaki et al. demonstrated that different functions describe different relaxation mechanisms occurring with different characteristic times. For example, the stretched exponential function used here was attributed to the motion of the collagen molecules (characteristic time equal to 25 – 45 minutes) and it did not describe well both the long-term bone relaxation, which was attributed to the viscous-like motion between cement lines (characteristic time above 70h) and the early relaxation response (~ 1 s from load application) (Goto et al., 1999; Nakayama et al., 1993). This observation indicates that while the base function used here is adaptable, as mentioned by the reviewer, the type of base function fitted to the data is also important in determining the quality of the fit. It should also be noted that regression methods are commonly used to test hypotheses on the underlying mechanism, through statistical inference, and their quality is to be considered in terms of the fraction of the variance in the data explained by the model and the statistical power for hypothesis testing, which is the approach used in the present study. The text now states (line 319):

“Fourthly, the single time point and the pixel size in the images (i.e., 30 microns) did not allow to capture the relative movement of collagen molecules thought to cause the fast relaxation process of bone. Nevertheless, the present study confirms the validity of the fast relaxation model developed earlier by Sasaki and co-workers [9] and provides the characteristic time and the shape factor for describing the force relaxation of human femurs.”

22. Line 257-258: *A must be related to the peak force as it is an amplitude whereas the exponential represents a ‘damper’.*

There is no constraint in the fitting function ensuring that A matches the force at t_0 . Rather, the goodness of the match depends on the number of points, the noise on the data, the base function fitted to the data, and the searching algorithm for the best fit solution. Therefore, A is a result and may be seen as a local indicator of fitting quality. The text now states (line 196):

“The goodness of fit of the stretched exponential function was assessed by calculating the coefficient of determination and by comparing the initial force F_0 and the scaling factor A.”

23. Line 265: *The separation into groups below and above a force ratio of 0.4 is not clear. I am aware of the answer to previous reviewers but this rationale should be made clear in the manuscript including the motivation and data for such a split.*

Thank you for this comment. Comparing the relaxation response at low and high loads, which cause minimal and sub-maximal levels of damage, can provide quantitative information about how much the damage process affects relaxation. This is the rationale behind the separation into two groups, below and above a force ratio of 0.4. Please refer to the author's answer to comment 12 for details about how this point has been clarified in the text.

24. Figure 5: *I am not sure what Fig 5 adds to the results at $t = 0$, the exponent becomes 1 and, thus, $A = F(0)$ according to eqn (1).*

Please refer to the answer to comment 22.

25. Line 287: *change "white women donors" to "white female donors"*

Done. Thank you.

Reviewer: 2

GENERAL COMMENTS

26. *This paper presents a re-analysis of previously published experiments. This causes some major limitations of this manuscript:*

The aim itself is not quite clear. What does the author ultimately aim to demonstrate?

This comment contrast with the comment by Reviewer #1, who commends the manuscript for the relevance of the study and the clarity of the aims. Reviewer #1 states:

"There is certainly novelty in the performing relaxation tests on whole femurs and analysing the results with regards to donor age and BMD. [...] The manuscript is very well written the aims of the study are clear and hypotheses are stated."

While the main text states (line 33):

"The aim of the present study was 1) to test the validity of earlier bone stress relaxation theories developed for bone cores for predicting the force relaxation response of human femora and 2) to falsify the hypothesis that aging modulates the femur's relaxation response."

27. B) *The experimental setup and protocol were clearly not designed to assess bone relaxation. Indeed, the force relaxation observed is the sum of several contributions:*

- *The bone, sure;*
- *The cartilage, which is definitely viscoelastic;*
- *The polyethylene cup used to deliver the force to the head;*
- *The acrylic cement used to constrain the distal end (due to the long lever arm, a small deformation of the cement can result in a relatively large deflection of the head.*

The reviewer is correct in that the experiments were not originally designed to assess bone relaxation, but this does not mean that the measurements do not represent the force relaxation response of the bone specimen.

The total displacement (deformation) is the sum of the displacement (deformation) of the different parts, not the force, which is equally applied to the pressure socket, the cartilage, the bone, and the distal constraint, for equilibrium. The cortical deformation changed on average by about 1% during the first-minute relaxation and was practically unchanged for the rest of the experiment (supplementary figures 1 and 2). The minimal change of cortical deformation was supported by the visual inspection of the images showing that every bone movement must have been smaller, or comparable, to the pixel size (30 microns) and much smaller than the displacement caused by the load (supplementary figure 3). This provides confidence in the validity of the relaxation experiment. Also, the reviewer does not realize that the viscoelastic time constant of the cartilage is in the tenth of seconds, interestingly similar to the minute within which most of the bone deformation changes occurred, so potentially affecting the bone relaxation response for a much smaller than the time interval analyzed here (30 minutes). Finally, the 1% change of bone deformation can be expected to result in an equivalent 1% change of the force measured due to the linearity ($R^2 > 0.95$) of the force-deformation response of bone (Dall'ara et al., 2013; Schileo et al., 2007). As such, the effect of the non-bone components was negligible. The text has been substantially amended to clarify how the results of this study provide confidence in the validity of the conclusions. The text now states (line 186):

“Data analysis

The consistency of the force relaxation measurements obtained using the two different testing devices and a variable number of imaging cycles was analyzed using robust linear regression. The change over time of bone deformation attributable to the non-bone components of the specimen assembly and the performance of the testing devices was assessed using both the cortical strain measurement and the images. The cortical strain measurements were preliminarily screened for outliers and six-order low-pass Butterworth filtered (50 Hz). Changes of the equivalent

cortical strain were calculated for the entire duration of the 30-minute relaxation experiment. A visual inspection of the images was conducted to ascertain no substantial movement, relative to the initial displacement, occurred during imaging (25.2 minutes) over the entire bone volume.”

And (line 207):

“Overall, 29 relaxation time histories were analyzed, 22 obtained with concomitant imaging data and 7 with no imaging data. One relaxation time history was discarded. By pooling all the 29 measurements, the force ratio (F_0/F_S) ranged between 7% and 78%. The compressive force decreased by an average 21% ($R_2 = 0.97$, $p < 0.01$) after 5 minutes and by 36% ($R_2 = 0.91$, $p < 0.01$) after 30 minutes. The force fraction lost at 30 minutes, FFL, was positively associated with the force ratio ($R_2 = 0.62$, $p < 0.01$), showing an average 7.7% increase in response to a 10% increase of the force ratio (Figure 2). The cortical strain measurements changed, on average, by less than 1% during the first minute of the relaxation experiment and remained substantially constant for the remaining 29 minutes (Supplementary figures 2 and 3). The entire volume of all the specimens displayed no substantial motion artifacts in the images (Supplementary figure 4).”

And (line 279):

“The strong correlation between the initial force and the force decline after 5 and 30 minutes ($R_2 > 0.91$) across all the force recordings obtained with different devices and variable imaging cycles provided confidence in the consistency of the data. Furthermore, the modest 1 % change of cortical strain during the first-minute relaxation is consistent with no substantial motion visible in the images across the entire bone volume and interestingly comparable to the characteristic relaxation time of the cartilage [30]. These modest bone strain changes are expected to cause similar changes in the force response of the bone specimen due to the linearity of the bone’s force-displacement relationship [31], hence indicating that most of the force relaxation reported here can be attributed to the bone while the cartilage may have played a role in the initial small drift of bone strain.”

The lever arm was minimal since the loading axis was designed to pass through the hip center and the distal cross-section of the femur facing the aluminum cup. A figure has been added to clarify this point (supplementary figure 1).

28. Due to the experimental design, it is not possible to ascertain the weight of the different components.

Please refer to the answer to the previous comment of the reviewer providing an in-depth analysis of the results and their validity.

29. *For instance, if the cartilage relaxed some tenths of millimeter, this was probably sufficient to cause a drop of the force of some hundred N; as bone tissue is quite stiff, this is likely to result in a small variation of the strain measured by the strain gauges. Indeed, a steady trend of the strain variation is reported in Supplementary Fig. 2.*

Bone responds linearly to changes in deformation so the 1% change in bone deformation produced a corresponding 1% change of force, limited to the first-minute relaxation. Please refer to the above answer 28 to the reviewer for a detailed explanation about how this point was included in the text.

30. *C) Data come from two test campaigns:*

- In one session, the bone was additionally exposed to large amount of X-ray radiation

The manuscript has been re-organized to directly link the results with the analysis of the quality of the data, including the effect of different testing modalities and the compression scheme. Please refer to the author's answer to comment 27 for details of how the text has been amended to quantify the consistency of the data.

31. *Was the polyethylene cup used in the two campaigns identical?*

Yes, it was. This detail is now reported in figure 1 and the text (line 209), which now states:

"In summary, the specimen was compressed via a spherically shaped polyethylene pressure socket of similar shape and stiffness to the natural acetabulum (Figure 1)."

32. *Was the cement constraint in the two campaigns identical?*

No, it was the same cement constraint mounted to both devices. This detail is now included in the text (line 205):

"The same specimen assembly (i.e., the specimen, the cement, and the aluminum cup) and same single-leg stance loading configuration were used in every test."

33. *These are additional confounding factors that cannot be ignored, but it is not possible to ascertain their contribution.*

For these reasons, I am afraid that the results and conclusions are not sufficiently robust to grant publication.

The reviewer may reconsider his/her opinion in light of the explanations provided.

BIBLIOGRAPHY

- Bowman, S.M., Gibson, L.J., Hayes, W.C., McMahon, T.A., 1999. Results from demineralized bone creep tests suggest that collagen is responsible for the creep behavior of bone. *J. Biomech. Eng.* 121, 253–258. <https://doi.org/10.1115/1.2835112>
- Dall'ara, E., Luisier, B., Schmidt, R., Kainberger, F., Zysset, P., Pahr, D., 2013. A nonlinear QCT-based finite element model validation study for the human femur tested in two configurations in vitro. *Bone* 52, 27–38. <https://doi.org/10.1016/j.bone.2012.09.006>
- Goto, T., Sasaki, N., Hikichi, K., 1999. Early stage-stress relaxation in compact bone. *J. Biomech.* 32, 93–97. [https://doi.org/10.1016/S0021-9290\(98\)00138-9](https://doi.org/10.1016/S0021-9290(98)00138-9)
- Jackman, T.M., Hussein, A.I., Curtiss, C., Fein, P.M., Camp, A., De Barros, L., Morgan, E.F., 2016. Quantitative, 3D Visualization of the Initiation and Progression of Vertebral Fractures under Compression and Anterior Flexion. *J. Bone Miner. Res.* 31, 777–788. <https://doi.org/10.1002/jbmr.2749>
- Lakes, R., Saha, S., 1979. Cement line motion in bone. *Science* (80-.). 204, 501–03. <https://doi.org/10.1126/science.432653>
- Nakayama, Y., Yoshikawa, M., Sasaki, N., Nakayama, Y., Yoshikawa, M., Enyo, A., 1993. Stress relaxation function of bone and bone collagen. *J. Biomech.* 26, 1369–1376. [https://doi.org/10.1016/0021-9290\(93\)90088-V](https://doi.org/10.1016/0021-9290(93)90088-V)
- Schileo, E., Taddei, F., Malandrino, A., Cristofolini, L., Viceconti, M., 2007. Subject-specific finite element models can accurately predict strain levels in long bones. *J. Biomech.* 40, 2982–2989. <https://doi.org/10.1016/j.jbiomech.2007.02.010>

Appendix B

RESPONSE TO THE REVIEWER COMMENTS

THE EFFECT OF AGE AND INITIAL COMPRESSION ON THE FORCE RELAXATION RESPONSE OF THE FEMUR IN ELDERLY WOMEN

Saulo Martelli

Reviewer #2 is correct in that the specific comments he/she provided were not answered in the previous response. This was not intentional and likely originated at the time of copying the reviewer comments from the Journal's email to the response document. Apologies for the confusion. Reviewer #2 also remarks general aspects, already discussed in the first review cycle, about the two experimental protocols, the effect of radiation, and the statistical analysis. The consistency of the two experimental protocols and the effect of radiation are here discussed further. The statistical analysis of the effect of age has been expanded, by sub-grouping, to address the problem of some specimens tested multiple times. The present response provides a point-by-point response to both reviewers, including the specific comments by reviewer #2, not addressed earlier.

Reviewer: 2

It is difficult to follow the author's responses as numbering and lettering of the points are mixed (e.g., my first general comment, which I had labelled as "A" is listed just as "26", my former point "B" has become "27 B" etc).

In the previous review, the author's responses were continually numbered, not mixed, so to help cross-referencing within and across reviewers. The structure of the response was described in the opening of the document, aiming at improving clarity, not at making the review process more difficult. In the present response, the author responses are not numbered and differentiated only by the text style to address the reviewer difficulty in following the previous structure. The reviewer comments are reported in italics and the author's response is given immediately below in plain text.

Additionally, it seems that the authors have not considered nor responded to the specific comments on the previous version. Therefore, I am not going to waste more time writing more specific comments, I will just focus on the main ones

The Author should have checked more carefully, before submitting.

Noted. This was not intentional and likely originated at the time of copying the reviewer comments from the Journal's email to the response document. Apologies for the confusion.

The Author stresses the different views of the two Reviewers. The fact that two Reviewers have different opinions is not uncommon (for instance, because they have different backgrounds), and does not undermine the relevance of the respective comments.

The reviewer is correct in that it is common for reviewers to have different, equally valid, opinions. The contrast highlighted in the previous review aimed at justifying the choice of not changing the aim statement, as an equally valid solution, not at undermining the relevance of one comment over another. The comments of the different reviewers have always been considered equally, here and across the entire text.

A) In the view of this Reviewer, the aim itself is not quite clear. It is true that two aims are stated. What is not clear is how this relates to the title (where strength is mentioned). This problem remains in the revised paper, where this problem is unsolved. Also, my specific comment #2 to the previous version related to the definition of the aims of this study.

Thank you for clarifying. The title was related to aim 2 where the moderate effect of age was considered in contrast with the known strong effect of age on BMD and fracture load. The title has been re-written focusing on what was done rather than on what was found. The title now states:

“THE EFFECT OF AGE AND COMPRESSIVE FORCE ON THE FORCE RELAXATION OF THE FEMUR IN ELDERLY WOMEN”

The reviewer specific comment #2 is now addressed in the end of the response to the reviewer #2.

Moreover, for what concerns Question 2: a hypothesis can be falsified by a positive finding. In your case, your study only found a correlation that was not significant. This could be due to a true lack of correlation, but also to having tested a very small number of specimens, covering a narrow range of ages of post-menopausal women.

The BMD in the 12 specimens declined with age by 0.01 g/cm² per year ($R^2 = 0.27$, $p = 0.05$) in agreement with known trends in the population [32,33] while the age range (56 – 91) was more than three decades wide, representing more than 40% of the expected lifespan for advanced societies. This information, reported in the second paragraph of the results section,

evidence that sample size, and the age range, provided a meaningful representation of the natural age-related bone decline.

An additional analysis expanded the previous analysis of the effect of age on force relaxation, by sub-grouping, and showing that age is moderately associated to the characteristic time when the initial compression is restricted to a narrow range of initial compression, although the association disappears when the data were all pooled together or grouped by excluding the force profiles with an initial compressive force higher than the 40% of the expected fracture load. This is now reported in the main text at lines 257 – 262:

“The logarithm of the characteristic time, τ , increased with age ($R^2 = 0.37$, $p = 0.03$) by pooling one force profile per specimen spanning a narrow range of the initial compression ($F_0/F_L = 0.23 - 0.43$; $n = 11$), explaining the 35% of the variation of the logarithm of the characteristic time in the entire dataset. No association was found between age, scaling factor, A , shape factor, β , and characteristic time, τ , either in the whole dataset or by excluding the force profiles with the highest initial compression ($F_0/F_L > 0.4$).”

Therefore, the association between the initial compression and relaxation was much stronger than that between age and relaxation. This is the main result of the analysis reported and clarified further in the conclusion paragraph, lines 334 – 342, and the abstract.

B) The answer to my concern about confounding factors is not convincing.

It is obvious that “total displacement (deformation) is the sum of the displacement (deformation) of the different parts, not the force, which is equally applied to the pressure socket, the cartilage, the bone, and the distal constraint, for equilibrium”. Exactly for this reason: if any of the components (the bone and/or the cement and/or the socket and/or the cartilage relaxates, you will see a drop of the force.

If it is obvious that the force does represent the boundary condition of the bone, then force applied to each component cannot change independently (answer #27 of the former review) and the experiment is a valid relaxation experiment if the deformation of the bone remains constant throughout the experiment. In other words, the effect of all the confounding factors mentioned by the reviewer on bone relaxation can be quantified by monitoring the deformation of the bone. This was the only purpose of the images and the cortical strain measurement showing that the drop of deformation is 1% in average, limited to the first 1 – 2 minutes of relaxation, necessarily causing a similar drop of the force from the ideal relaxation conditions. The description of the analysis (lines 182 – 192), the results section (first

paragraph) and the discussion (first and second paragraphs) have been edited further to clarify this point.

C) Data come from two test campaigns. The experimental setup and protocol were clearly not designed to assess bone relaxation as one has to move through datasets and assumptions to compare data that are not comparable: Some specimens were tested multiple times (hence have a different weight in the stats), some underwent massive x-radiation, strain was measured, but only in a subset of specimens, etc. (even the reply about the cement is unclear: "was the cement identical?" "No, it was the same cement constraint mounted to both devices"... and then line 205 does not mention the cement

These are additional confounding factors that cannot be ignored, but it is not possible to ascertain their contribution.

Regarding the multiple imaging cycles, the previous review (answer #8 and #27) clarified that 1) the present protocol did not allow the full radiation to reach the specimen, 2) that earlier mechanical studies of entire bones showed no effect of x-ray radiation on bone integrity, in apparent contrast to the reported compromised integrity reported earlier by radiating small bone cores, and 3) that the initial force reported here (Figure 2) explains 97% of force variation at 5 minutes, and 91% at 30 minutes, irrespective of the testing device and number of imaging cycles. In the current revision, the residuals of the regression between the initial compression and the force recorded at 5- and 30-minutes reveals no differences (two-sample t-test, $p > 0.45$) between the force profiles recorded using the two different testing devices and a different number of imaging cycles (lines 189 – 191; 208 – 210; 279 - 282).

Regarding the ability of the two different devices, the 1% change of bone deformation in the first 1 – 2 minutes relaxation provides a measure of the deviation from the ideal relaxation condition attributable to the properties of all the non-bone components in the specimen assembly and the performance of the two devices. The main text was amended to clarify this aspect as described in the answer to comment B.

Finally, the data have been analysed further by sub-grouping to address the variable number of tests per specimen and the large variation of the initial compression. Using one test per specimen and a narrow initial compression ($n = 11$, $F_0/FL = 0.29 \pm 0.7$, $CV = 24\%$), age and initial compression were both moderately associated with the logarithm of the characteristic time ($R^2 = 0.37$ and $R^2 = 0.50$), where age alone explained 35% of the variation in the whole dataset of the variation of the logarithm of the characteristic time. However, the

effect of age was not detected by pooling all the force profiles below 40% of the expected fracture load, nor by pooling all the data ($F_0/F_L < 0.78$). Therefore, the effect of age is smaller than that of the initial compression. This is now clarified in the main text (conclusions, lines 334 – 342, 256 – 262, and abstract)

The response on the cement was reported in the main text of the former review at line 105, not 205, due to a typo. However, the sentence of the main text was copied immediately below for easy access during the review.

D) Methods are presented in a very confusing way, for instance:

- To understand the actual force values delivered to the different specimens one has to gather this information from the text, plus the tables, plus reading another manuscript.

- Knowing the force value and the force ratio would help the reader understand, for instance, if the two experiments (the radiation-exposed one and the robot one) overlapped in terms of force ranges. This is fundamental, for instance, to interpret Fig. 4 (formerly fig. 3)

- What is the logical order of the rows in Table 2?

The initial force values are all reported in table 2 and the entire force time histories for both the experiment modalities are separately accessible for review at <https://figshare.com/s/2a7074c292441638a7f7>. This is clearly indicated in the data accessibility statement and in accordance with the Journal policy and guidelines. The caption of table 2 has been re-written to better describe its content and to allow discriminating between the data obtained using the two experimental modalities.

SPECIFIC COMMENTS

Two line-numbering series appear, one from the Author, the other one by the system. The comments below refer to the line numbers next to the text.

The paper is generally well-written and the images are clear. Here are some detailed comments:

1. INTRODUCTION, line 74: Did they observe “no change”, or the changes were “not statistically significant”?

The differences were not statistically different. The text has been amended accordingly.

2. *INTRODUCTION, line 76-77: The present study does not really “test the validity of earlier theories”. What it does, is test their suitability to describe the behaviour of entire bone by extrapolating them from the tissue level.*

What it does is fitting earlier theories to the data and assess the goodness of fit. The term validity was not meant to refer to a validation study but rather to ascertain if the model could be used, as a valid option, for entire bones. The terms suitability appears indeed less ambiguous, and it is now used instead throughout the text.

3. *INTRODUCTION, line 85-86: I would assume that the ethics clearance belongs to the M&M, not to the intro.*

The ethics statement has been moved to the first line of the method section.

4. *MATERIALS AND METHODS, lines 130-131: “constant deformation” can be deceiving as it is possible that some components relaxed significantly (see major comments above). What was constant was the position of the two ends. We don’t know exactly what happened between them.*

As pointed out above, this is a point of disagreement because what happened to the bone has been, in fact, quantified.

5. *MATERIALS AND METHODS, lines 135: “minimal deformation changes” is not a scientific description. This should be reported in quantitative terms.*

The term has been removed.

6. *MATERIALS AND METHODS, lines 145: “appreciable movement artefacts” is not a scientific description. This should be reported in quantitative terms. It is possible that movements were satisfactorily small for the original study, but possibly too large for the purposes of the present analysis.*

The text has been amended accordingly and now it states:

“A visual inspection of the images was conducted to ascertain that every movement occurred during imaging (25.2 minutes) was comparable of below the pixel size (0.03 mm) over the entire bone volume.”

7. *MATERIALS AND METHODS, lines 219 (and in the entire paper, figures, tables): Is there a specific reason for calling “Fs” the failure force? It is not mnemonic, and can easily be misread as “F5” (which is also used in the paper).*

The symbol has been changed to FL (Fracture Load) to reduce ambiguity.

8. *RESULTS, line 242-243: you must provide an explanation of disregarding one specimen.*

That was due to data corruption. This information has been included in the opening of the results section.

9. *DISCUSSION and CONCLUSIONS: should be entirely revisited to account for the major limitations described above.*

The discussion and conclusion were extensively revised to address these points during the first revision and are here amended further. Changes of the main text are tracked and visible.

Reviewer: 1

Comments to the Author(s)

The responded to all my questions and comments. There are a couple of points that I consider under scientific debate which must respect different opinion and for which I am grateful.

The sticky point is that I still believe that this is largely an extended repetition of previous experiments [18, 19] and, thus, an overall incremental manuscript. This has also been flagged up by reviewer 2 and my comments in the initial review with regards to novel aspects of the paper do not serve to counter the comments of reviewer 2. This is, however, an editorial issue and not an issue of the overall content of the manuscript and the author's responses, which were written well.

Thank you for this comment. The data used in the present study are undoubtedly an extension of data collected earlier, using established technologies, aimed solely at reaching a sufficient sample size for a meaningful statistical analysis. I believe that this has always been clearly acknowledged in the text. The actual novelty of the manuscript is, in fact, the statistical analysis, the results and the conclusion. Whether the data, or the results, or both, should be used to define an extension study is certainly an interesting point of discussion likely best placed at the editorial level of the journal, as suggested by the reviewer.